# State-Covering Trajectory Stitching
# for Diffusion Planners

**Kyowoon Lee**
KAIST
leekwoon@kaist.ac.kr

**Jaesik Choi**
KAIST, INEEJI
jaesik.choi@kaist.ac.kr

## Abstract

Diffusion-based generative models are emerging as powerful tools for long-horizon planning in reinforcement learning (RL), particularly with offline datasets. However, their performance is fundamentally limited by the quality and diversity of training data. This often restricts their generalization to tasks outside their training distribution or longer planning horizons. To overcome this challenge, we propose *State-Covering Trajectory Stitching* (SCoTS), a novel reward-free trajectory augmentation method that incrementally stitches together short trajectory segments, systematically generating diverse and extended trajectories. SCoTS first learns a temporal distance-preserving latent representation that captures the underlying temporal structure of the environment, then iteratively stitches trajectory segments guided by directional exploration and novelty to effectively cover and expand this latent space. We demonstrate that SCoTS significantly improves the performance and generalization capabilities of diffusion planners on offline goal-conditioned benchmarks requiring stitching and long-horizon reasoning. Furthermore, augmented trajectories generated by SCoTS significantly improve the performance of widely used offline goal-conditioned RL algorithms across diverse environments. Our code is available at https://github.com/leekwoon/scots/

## 1 Introduction

In many real-world applications, agents must plan over hundreds of steps, often receiving sparse or delayed feedback until they reach a distant goal. Perfect knowledge of the environment allows powerful planners like MPC (Tassa et al., 2012) and MCTS (Silver et al., 2016, 2017) to excel. However, most real-world tasks instead require learning environment dynamics from data. Model-based reinforcement learning (MBRL) (Sutton, 2018) constructs such world models, offering sample-efficient learning and improved generalization (Ha & Schmidhuber, 2018; Hafner et al., 2019; Kaiser et al., 2020). However, autoregressive predictions from learned models accumulate small errors into a cascade of inaccuracies. This compounding error can cause planners to exploit model inaccuracies and generate trajectories that are suboptimal or even physically infeasible, especially in long-horizon tasks (Talvitie, 2014; Asadi et al., 2018; Janner et al., 2019; Voelcker et al., 2022; Chen et al., 2024a).

To address these limitations, diffusion planners (Janner et al., 2022; Ajay et al., 2023; Liang et al., 2023; Chen et al., 2024c) have recently emerged as a promising alternative for trajectory generation in sequential decision-making. Instead of rolling out one step at a time, diffusion planners treat each trajectory as a single high-dimensional sample, learning a denoising process that transforms noise drawn from a simple prior into trajectories that match the target distribution (Ho et al., 2020; Song et al., 2021). By operating on entire trajectories simultaneously, these methods inherently prevent the compounding of prediction errors that undermine autoregressive dynamics models. Moreover, the generative nature of diffusion models allows for flexible conditioning and guidance mechanisms, enabling the synthesis of plans with properties like reaching specific goals or maximizing expected returns (Dhariwal & Nichol, 2021).

39th Conference on Neural Information Processing Systems (NeurIPS 2025).

Despite these advantages, the effectiveness of diffusion planners remains fundamentally limited by the quality, diversity, and coverage of the offline training data. First, their effective planning horizon is inherently coupled to the maximum trajectory length observed during training, making it challenging to generate coherent plans that significantly exceed this length. Second, their generalization capability is often confined to the specific types of trajectories and transitions represented in the training data. For instance, if the dataset predominantly features certain movement patterns, the planner may struggle to synthesize solutions for novel tasks requiring different compositions of behaviors (as illustrated in Figure 1). While exhaustively collecting data for all conceivable scenarios could mitigate this, such an approach is prohibitively expensive. Trajectory stitching (Ziebart et al., 2008) offers a promising alternative by composing novel, longer sequences from existing short segments. However, existing methods rely heavily on extrinsic rewards for segment selection, and maintaining the dynamic consistency and feasibility of stitched trajectories remains challenging.

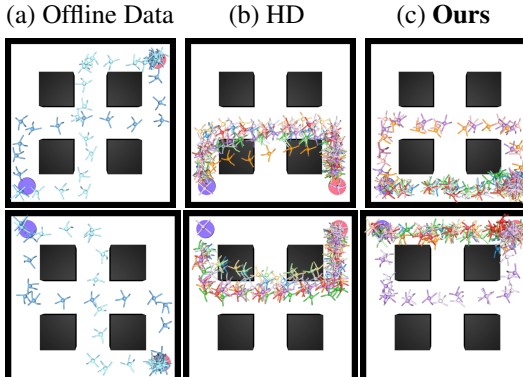

(a) Offline Data    (b) HD    (c) **Ours**

Figure 1: **Improved generalization with SCoTS.** (a) Examples from the training dataset, illustrating limited coverage. (b) Plans generated by Hierarchical Diffuser (HD) (Chen et al., 2024c), which fail to generalize well to these out-of-distribution tasks due to insufficient coverage of the training data. (c) Plans generated by HD trained on SCoTS-augmented data, demonstrating significantly improved trajectory stitching capability and generalization to unseen tasks. Each color corresponds to one of 10 plans generated by the planner.

In this paper, we propose **S**tate-**Co**vering **T**rajectory **S**titching (**SCoTS**), a reward-free trajectory augmentation framework that systematically extends trajectories to cover diverse, unexplored regions of the state space. Specifically, SCoTS employs a three-stage approach: First, we learn a temporal distance-preserving latent representation by training a model to encode states based on learned optimal temporal distances, facilitating efficient identification of viable trajectory segments. Second, we introduce a novel iterative stitching strategy that balances directed exploration with state-space coverage. In this process, trajectory segments are selected based on their progress along a learned direction in the latent space and their novelty relative to previously explored regions within the rollout. Finally, we refine the resulting stitched trajectories using a diffusion-based refinement procedure. Consequently, the resulting trajectories exhibit broader state-space coverage while preserving dynamic feasibility.

To summarize, our contribution in this paper is the introduction of SCoTS, a reward-free trajectory augmentation approach designed to generate diverse, long-horizon trajectories that enhance diffusion planners. Extensive experiments across diverse and challenging benchmark tasks show that SCoTS significantly enhances the stitching capabilities and long-horizon generalization of diffusion planners. Furthermore, augmented trajectories generated by SCoTS notably boost the performance of widely used offline goal-conditioned reinforcement learning (GCRL) algorithms in across multiple trajectory stitching benchmarks.

## 2 Planning with Diffusion Models

Diffusion-based planners (Janner et al., 2022; Liang et al., 2023; Chen et al., 2024c) provide a promising framework for long-horizon decision-making by modeling entire trajectories as joint distributions. A trajectory $\boldsymbol{\tau}$ is typically represented as a sequence of states $\boldsymbol{s}_t$ and actions $\boldsymbol{a}_t$ over a planning horizon $T$:

$$\boldsymbol{\tau} = \begin{bmatrix} \boldsymbol{s}_1 & \boldsymbol{s}_2 & \dots & \boldsymbol{s}_T \\ \boldsymbol{a}_1 & \boldsymbol{a}_2 & \dots & \boldsymbol{a}_T \end{bmatrix}, \tag{1}$$

where $\boldsymbol{s}_t$ and $\boldsymbol{a}_t$ denote the state and action at time step $t$, respectively. Diffusion planners utilize diffusion probabilistic models (Sohl-Dickstein et al., 2015; Ho et al., 2020) to learn a trajectory distribution $p_\theta(\boldsymbol{\tau}^0)$ over noise-free trajectories $\boldsymbol{\tau}^0$. This involves a predefined forward noising process

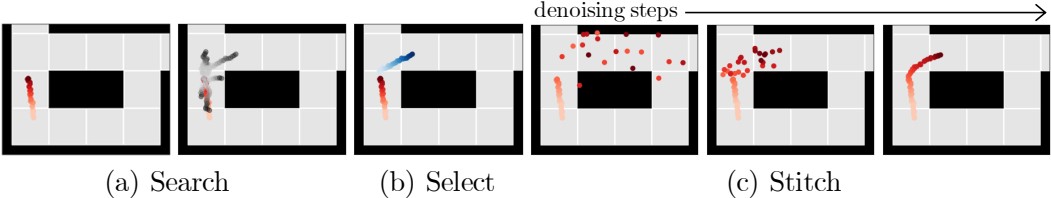

(a) Search        (b) Select        (c) Stitch

Figure 2: **Overview of the SCoTS stitching process.** (a) **Temporal Distance-Preserving Search:** Given the currently composed trajectory (red), we identify candidate segments (gray) by searching in a latent space learned to preserve temporal distances. Candidates are selected based on proximity to the endpoint of the current trajectory in latent space. (b) **Exploratory Segment Selection:** Among the retrieved candidate segments, we select the segment (blue) that best balances directional progress toward a randomly sampled latent direction and novelty relative to previously visited states in latent space. (c) **Diffusion-based Stitching Refinement:** To ensure smooth transitions, a diffusion model refines the stitching point between segments, generating dynamically consistent trajectories.

and a learned reverse denoising process. The forward process incrementally adds Gaussian noise to the trajectories through $M$ discrete diffusion timesteps with a variance schedule $\{\beta_i\}_{i=1}^{M}$:

$$q(\boldsymbol{\tau}^i|\boldsymbol{\tau}^{i-1}) := \mathcal{N}(\boldsymbol{\tau}^i; \sqrt{1-\beta_i}\boldsymbol{\tau}^{i-1}, \beta_i\mathbf{I}). \tag{2}$$

A key property is the direct sampling of intermediate trajectories:

$$q(\boldsymbol{\tau}^i \mid \boldsymbol{\tau}^0) = \mathcal{N}(\boldsymbol{\tau}^i; \sqrt{\alpha_i}\boldsymbol{\tau}^0, (1-\alpha_i)\mathbf{I}), \tag{3}$$

where $\alpha_i := \prod_{s=1}^{i}(1-\beta_s)$. The schedule ensures that $\boldsymbol{\tau}^M$ approximates a standard Gaussian distribution $\mathcal{N}(\mathbf{0}, \mathbf{I})$. The reverse process learns to invert this noising process and define following generative process with a standard Gaussian prior $p(\boldsymbol{\tau}^M)$:

$$p_\theta(\boldsymbol{\tau}^0) = \int p(\boldsymbol{\tau}^M)\prod_{i=1}^{M} p_\theta(\boldsymbol{\tau}^{i-1}|\boldsymbol{\tau}^i)\,\mathrm{d}\boldsymbol{\tau}^{1:M} \tag{4}$$

with a learnable Gaussian transition: $p_\theta(\boldsymbol{\tau}^{i-1}|\boldsymbol{\tau}^i) = \mathcal{N}(\boldsymbol{\tau}^{i-1}|\boldsymbol{\mu}_\theta(\boldsymbol{\tau}^i, i), \boldsymbol{\Sigma}^i)$.

Given an offline dataset $\mathcal{D}$, diffusion models in practice simplify training by parameterizing a noise-prediction network $\boldsymbol{\epsilon}_\theta$, trained to predict the noise $\boldsymbol{\epsilon}$ added during the forward process (Ho et al., 2020):

$$\mathcal{L}(\theta) := \mathbb{E}_{i,\boldsymbol{\epsilon},\boldsymbol{\tau}^0}[\|\boldsymbol{\epsilon} - \boldsymbol{\epsilon}_\theta(\boldsymbol{\tau}^i, i)\|^2], \tag{5}$$

where $i \in \{0, 1, ..., M\}$ is the diffusion timestep, $\boldsymbol{\epsilon} \sim \mathcal{N}(\mathbf{0}, \mathbf{I})$ is target noise that was used to corrupt clean trajectory $\boldsymbol{\tau}^0$ into $\boldsymbol{\tau}^i = \sqrt{\alpha_i}\boldsymbol{\tau}^0 + \sqrt{1-\alpha_i}\boldsymbol{\epsilon}$.

**Remark.** Previous works generally assume the offline dataset $\mathcal{D}$ sufficiently covers diverse trajectories with substantial length. Consequently, these studies have primarily focused on improving network architectures, action generation methods, and planning strategies. In contrast, we explicitly aim to generate an augmented dataset $\mathcal{D}_{\text{aug}}$ that extends trajectory coverage, enabling diffusion planners to generalize effectively beyond their training distribution.

## 3 State-Covering Trajectory Stitching

We introduce **St**ate-**Co**vering **T**rajectory **S**titching (**SCoTS**), a novel *reward-free* trajectory augmentation framework designed to synthesize an augmented dataset $\mathcal{D}_{\text{aug}}$ from an offline dataset $\mathcal{D}$. The core idea of SCoTS is to iteratively construct long and diverse trajectories by repeatedly stitching short segments guided by latent directional exploration, resulting in significantly improved generalization and extended planning horizons for diffusion planners. SCoTS consists of three stages: (1) learning a temporal distance-preserving embedding for efficient segment retrieval (Section 3.1); (2) iterative trajectory stitching driven by latent directional exploration and novelty-based selection (Section 3.2); and (3) diffusion-based refinement to ensure dynamically consistent transitions (Section 3.3). The overall procedure of SCoTS, including segment search, exploratory selection, and diffusion-based refinement, is illustrated in Figure 2. The detailed algorithm is summarized in Algorithm 1.

**Algorithm 1 Overview of the SCoTS Framework**

1: **Input:** Offline dataset $\mathcal{D}$, Temporal distance-preserving embedding $\phi$, Diffusion stitcher $p_\theta^{\text{stitcher}}$
2: **Initialize:** Augmented dataset $\mathcal{D}_{\text{aug}} = \emptyset$
3: **for** $n = 1, \ldots, N_{\text{traj}}$ **do**
4:     // Sample initial segment from offline data
5:     $\boldsymbol{\tau}_{\text{comp}} \sim \mathcal{D}$
6:     // Sample a random latent exploration direction
7:     $\boldsymbol{z} \sim \mathcal{N}(\mathbf{0}, \mathbf{I}); \quad \boldsymbol{z} \leftarrow \boldsymbol{z}/\|\boldsymbol{z}\|$
8:     **for** $t = 1, \ldots, N_{\text{stitch}}$ **do**
9:         // Retrieve nearest segments using temporal embedding
10:        $\{\boldsymbol{\tau}_j\}_{j=1}^k \leftarrow \text{TopKNeighbors}(\phi(\text{end}(\boldsymbol{\tau}_{\text{comp}})), \phi(\mathcal{D}), k)$
11:        // Compute directional progress and novelty scores
12:        Compute scores $S_j = P_j + \beta N_j$                                    (Eq. (11))
13:        // Select best candidate segment
14:        $\boldsymbol{\tau}_{\text{best}} \leftarrow \arg\max_j S_j$
15:        // Diffusion-based stitching refinement
16:        $\boldsymbol{\tau}' \sim p_\theta^{\text{stitcher}}(\cdot \mid \boldsymbol{s}_1 = \text{end}(\boldsymbol{\tau}_{\text{comp}}), \boldsymbol{s}_H = \text{end}(\boldsymbol{\tau}_{\text{best}}))$
17:        // Concatenate refined segment to trajectory
18:        $\boldsymbol{\tau}_{\text{comp}} \leftarrow [\boldsymbol{\tau}_{\text{comp}}, \boldsymbol{\tau}']$
19:    **end for**
20:    $\mathcal{D}_{\text{aug}} \leftarrow \mathcal{D}_{\text{aug}} \cup \{\boldsymbol{\tau}_{\text{comp}}\}$
21: **end for**
22: Train diffusion planner on $\mathcal{D}_{\text{aug}}$

## 3.1 Temporal Distance-Preserving Embedding

Identifying trajectory segments that are suitable for stitching requires accurately measuring their temporal closeness. However, simply using raw state-space distances can yield temporally incoherent results due to potential dynamic inconsistencies arising from ignoring state reachability. To address this, we employ a temporal distance-preserving embedding $\phi : \mathcal{S} \to \mathcal{Z}$, which maps raw states to a latent space $\mathcal{Z}$ designed such that the Euclidean distance $\|\phi(\boldsymbol{s}) - \phi(\boldsymbol{g})\|_2$ approximates the optimal temporal distance $d^*(\boldsymbol{s}, \boldsymbol{g})$, defined as the minimum number of environment steps required to transition from state $\boldsymbol{s}$ to state $\boldsymbol{g}$. Formally, we parameterize a goal-conditioned value function $V(\boldsymbol{s}, \boldsymbol{g})$ following (Park et al., 2024a):

$$V(\boldsymbol{s}, \boldsymbol{g}) := -\|\phi(\boldsymbol{s}) - \phi(\boldsymbol{g})\|_2, \tag{6}$$

which is trained on the offline dataset $\mathcal{D}$ using a temporal difference objective inspired by implicit Q-learning (Kostrikov et al., 2022):

$$\mathcal{L}_\phi := \mathbb{E}_{(\boldsymbol{s}, \boldsymbol{a}, \boldsymbol{s}', \boldsymbol{g}) \sim \mathcal{D}} \left[ \ell_\xi^2 (-\mathbb{1}(\boldsymbol{s} \neq \boldsymbol{g}) - \gamma\|\bar{\phi}(\boldsymbol{s}') - \bar{\phi}(\boldsymbol{g})\|_2 + \|\phi(\boldsymbol{s}) - \phi(\boldsymbol{g})\|_2) \right], \tag{7}$$

where $\bar{\phi}$ is a target network (Mnih, 2013), $\gamma$ is a discount factor, and $\ell_\xi^2$ denotes the expectile loss (Kostrikov et al., 2022; Newey & Powell, 1987).

**Remark.** We note that the learned latent space is not a perfect metric representation of the MDP (Park et al., 2024a). However, the reliability of SCoTS is grounded in its design, which does not require a globally accurate temporal distance metric. Instead, as we detail next, our framework leverages the latent distance for the more tractable local problem of retrieving promising and reachable candidate segments at each step of the iterative stitching process. This local approach makes the overall framework robust to the inherent imperfections of the embedding.

## 3.2 Directional and Exploratory Trajectory Stitching

Given the learned temporal distance-preserving embedding $\phi$, we iteratively construct extended trajectories via stitching. We start each new trajectory by randomly sampling an initial segment $\boldsymbol{\tau}_{\text{init}}$ from the offline dataset $\mathcal{D}$. To encourage diverse state coverage, we randomly sample a fixed latent exploration direction $\boldsymbol{z}$ as a unit vector, i.e., $\boldsymbol{z} \sim \mathcal{N}(\mathbf{0}, \mathbf{I})$, $\boldsymbol{z} \leftarrow \boldsymbol{z}/\|\boldsymbol{z}\|$, for each trajectory rollout.

At each stitching iteration, let $\boldsymbol{\tau}_{\text{comp}}$ denote the currently composed trajectory. We define $\text{end}(\boldsymbol{\tau})$ as a function returning the final state of trajectory $\boldsymbol{\tau}$. We then identify a set of candidate segments $\{\boldsymbol{\tau}_j\}_{j=1}^k$ whose initial states are nearest neighbors to $\text{end}(\boldsymbol{\tau}_{\text{comp}})$ in the latent space:

$$\{\boldsymbol{\tau}_j\}_{j=1}^k = \text{TopKNeighbors}(\phi(\text{end}(\boldsymbol{\tau}_{\text{comp}})), \phi(\mathcal{D}), k), \tag{8}$$

where $\phi(\mathcal{D})$ is a concise representation for the set of latent embeddings of the initial states of all trajectories within the dataset $\mathcal{D}$, and the distance metric is $\|\phi(\text{end}(\boldsymbol{\tau}_{\text{comp}})) - \phi(\boldsymbol{s}_{1,j})\|_2$.

To select the best candidate for stitching, we evaluate each candidate segment $\boldsymbol{\tau}_j = (\boldsymbol{s}_{1,j}, \ldots, \boldsymbol{s}_{H,j})$ based on a composite score balancing directional progress and novelty. The *progress score* quantifies the alignment in the latent space between the segment direction and the exploration direction $\boldsymbol{z}$:

$$P_j = \langle \phi(\text{end}(\boldsymbol{\tau}_j)) - \phi(\boldsymbol{s}_{1,j}), \boldsymbol{z} \rangle. \tag{9}$$

The *novelty score* promotes exploration and coverage of novel latent states by estimating the entropy of the endpoint of each candidate segment $\boldsymbol{\tau}_j$ relative to previously visited latent states. Here, $\mathcal{V}_{\text{rollout}}$ denotes the collection of latent representations of every state along previously stitched segments. Leveraging a non-parametric particle-based estimator (Liu & Abbeel, 2021) on our temporal distance-preserving embeddings, we compute the novelty score as:

$$N_j = \frac{1}{k_{\text{density}}} \sum_{\phi_v \in \text{k-NN}\left(\phi(\text{end}(\boldsymbol{\tau}_j)), \mathcal{V}_{\text{rollout}}, k_{\text{density}}\right)} \left\|\phi(\text{end}(\boldsymbol{\tau}_j)) - \phi_v\right\|_2. \tag{10}$$

A higher $N_j$ indicates greater novelty, signaling that the candidate segment expands coverage by moving towards less-explored regions of the latent space. We combine these two metrics to form the overall selection criterion:

$$S_j = P_j + \beta N_j, \tag{11}$$

where $\beta$ balances progress and novelty. We then stitch the candidate $\boldsymbol{\tau}_{\text{best}}$ with the highest score to $\boldsymbol{\tau}_{\text{comp}}$.

### 3.3 Diffusion-based Stitching Refinement

Although the exploratory selection step identifies segments with desirable progress and novelty, the stitching points, i.e., the connecting states between consecutive trajectory segments, may still exhibit minor dynamic inconsistencies or sub-optimal transitions. To mitigate these issues, we introduce a diffusion-based refinement step. Specifically, we train a diffusion model, termed the *stitcher* $p_\theta^{\text{stitcher}}$, which generates intermediate states conditioned on the boundary states of adjacent segments. Given a selected segment $\boldsymbol{\tau}_{\text{best}}$, the stitcher produces a refined trajectory $\boldsymbol{\tau}'$ by sampling from:

$$\boldsymbol{\tau}' \sim p_\theta^{\text{stitcher}}(\cdot \mid \boldsymbol{s}_1 = \text{end}(\boldsymbol{\tau}_{\text{comp}}), \boldsymbol{s}_H = \text{end}(\boldsymbol{\tau}_{\text{best}})), \tag{12}$$

where $\text{end}(\boldsymbol{\tau}_{\text{comp}})$ denotes the end state of the current composite trajectory $\boldsymbol{\tau}_{\text{comp}}$, and $\text{end}(\boldsymbol{\tau}_{\text{best}})$ denotes the end state of the newly selected segment $\boldsymbol{\tau}_{\text{best}}$. This diffusion-based refinement effectively smooths out transitions, ensuring dynamic coherence and feasibility of the stitched trajectories.

By iteratively repeating segment search, exploratory selection, and this refinement process, we construct a diverse set of augmented trajectories. To generate corresponding action sequences for these trajectories, we train an inverse dynamics model $\boldsymbol{a}_t = f_\psi(\boldsymbol{s}_t, \boldsymbol{s}_{t+1})$ on the offline dataset $\mathcal{D}$, which infers the actions that transition between consecutive states. The resulting state-action trajectories are aggregated into the augmented dataset $\mathcal{D}_{\text{aug}}$. This systematic and iterative augmentation approach generates an augmented dataset that broadly covers the state space. Crucially, diffusion planners trained on this augmented data exhibit significantly enhanced trajectory stitching capabilities and improved long-horizon generalization, particularly for tasks requiring extensive trajectory stitching and long-horizon reasoning (Section 4.3).

## 4 Experiments

In this section, we empirically validate the effectiveness of our proposed SCoTS framework. Specifically, we aim to investigate **(1)** whether SCoTS can generate diverse trajectories that extend significantly beyond the planning horizons present in the original offline dataset, **(2)** whether training

diffusion planners on these augmented trajectories enhances their capability to produce feasible long-horizon plans in unseen scenarios, and **(3)** whether the augmented dataset generated by SCoTS provides significant performance improvements for existing offline goal-conditioned reinforcement learning (GCRL) algorithms. Additional results can be found in Appendix C.

## 4.1 Datasets and Environments

We evaluate SCoTS on OGBench benchmark (Park et al., 2025), spanning diverse difficulties, environment sizes, agent state dimensions, and training data qualities. Specifically, the benchmark includes three locomotion environments: `PointMaze` (controlling a 2D point mass) and `AntMaze` (controlling an 8-DoF quadrupedal Ant). We consider two distinct dataset types, each designed to evaluate specific challenges. The `Stitch` dataset comprises short, goal-reaching trajectories limited to four cell units, thus requiring the agent to stitch multiple segments (up to 8) for successful inference. In contrast, the `Explore` dataset assesses learning navigation behaviors from extensive yet low-quality exploratory trajectories, collected by frequently resampling random directions and injecting significant action noise. For each environment, we report the success rate averaged over all evaluation episodes, where an episode is considered successful if the agent reaches sufficiently close to the goal state within a predefined distance threshold. See Appendix A for dataset details.

## 4.2 Diversity and State Coverage Analysis

To investigate whether SCoTS effectively promotes diverse state-space coverage through trajectory stitching, we evaluate its performance in the `PointMaze-Giant-Stitch` environment. As illustrated in Figure 3, we visualize the incremental stitching process for different values of the novelty weighting parameter $\beta \in \{0, 2, 20\}$. We observe that when $\beta = 0$, trajectory stitching predominantly follows latent directional guidance, resulting in trajectories with limited coverage but clear directional distinctions. With a moderate setting $\beta = 2$, trajectories exhibit a balanced trade-off, achieving substantial state-space coverage with notable diversity. Conversely, at a higher novelty weight $\beta = 20$, trajectories broadly cover the state space but lose their distinctiveness, leading to overlapping paths across different latent exploration directions. Based on these results, we use $\beta = 2$ across all environments in our experiments.

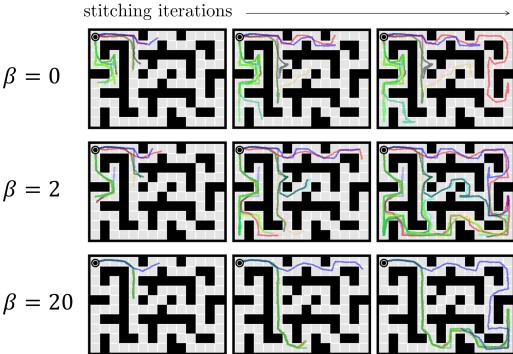

Figure 3: **Effect of novelty score on Trajectory Stitching.** Trajectory stitching examples in the `PointMaze-Giant-Stitch` environment. The original dataset (`Stitch`) consists of short segments limited to at most four maze cells. Different colors represent trajectories generated from distinct latent exploration directions $z$.

## 4.3 Diffusion Planning with SCoTS-Augmented Data

We next demonstrate how SCoTS-generated trajectories enhance the ability of diffusion planners to generate feasible, long-horizon plans beyond their training distribution. We compare our approach with offline goal-conditioned reinforcement learning (GCRL) methods including goal-conditioned implicit Q-learning (GCIQL) (Kostrikov et al., 2022), Quasimetric RL (QRL) (Wang et al., 2023), Contrastive RL (CRL) (Eysenbach et al., 2022), and Hierarchical implicit Q-learning (HIQL) (Park et al., 2023). We also include diffusion-based generative planning baselines explicitly designed for long-horizon generalization, such as Generative Skill Chaining (GSC) (Mishra et al., 2023) and Compositional Diffuser (CD) (Luo et al., 2025).

For our experiments, we adopt a hierarchical diffusion planner (HD) (Chen et al., 2024c) that generates plans through a two-level planning process. Specifically, the high-level diffusion model first generates sparse, temporally coarse waypoints, after which a low-level diffusion model fills in the intermediate states between these waypoints, producing a temporally dense trajectory. Initially constrained by limited and short-horizon training data, we augment the original dataset with SCoTS-generated trajectories. After dataset augmentation, we train diffusion planner and employ a value-based low-level controller for action execution, following recent approaches (Yoon et al., 2025a; Lu et al.,

Table 1: **Quantitative results on locomotion tasks in OGBench.** Results are averaged over 5 random seeds, each with 50 episodes per task. Standard deviations are reported after the $\pm$ sign.

| Env | Type | Size | GCIQL | QRL | CRL | HIQL | GSC | CD | HD | SCoTS |
|---|---|---|---|---|---|---|---|---|---|---|
| PointMaze | Stitch | Medium | 21 $\pm$9 | 80 $\pm$12 | 0 $\pm$1 | 74 $\pm$6 | 100$\pm$0 | 100$\pm$0 | 24$\pm$3 | 100$\pm$0 |
| | | Large | 31 $\pm$2 | 84 $\pm$15 | 0 $\pm$0 | 13 $\pm$6 | 100$\pm$0 | 100$\pm$0 | 17$\pm$2 | 100$\pm$0 |
| | | Giant | 0 $\pm$0 | 50 $\pm$8 | 0 $\pm$0 | 0 $\pm$0 | 29$\pm$3 | 68$\pm$3 | 0$\pm$0 | 100$\pm$0 |
| AntMaze | Stitch | Medium | 29 $\pm$6 | 59 $\pm$7 | 53 $\pm$6 | 94 $\pm$1 | 97$\pm$2 | 96$\pm$2 | 71$\pm$1 | 97$\pm$1 |
| | | Large | 7 $\pm$2 | 18 $\pm$2 | 11 $\pm$2 | 67 $\pm$5 | 66$\pm$2 | 86$\pm$2 | 36$\pm$2 | 93$\pm$1 |
| | | Giant | 0 $\pm$0 | 0 $\pm$0 | 0 $\pm$0 | 2 $\pm$2 | 20$\pm$1 | 65$\pm$3 | 0$\pm$0 | 87$\pm$2 |
| | Explore | Medium | 13 $\pm$2 | 1 $\pm$1 | 3 $\pm$2 | 37 $\pm$10 | 90$\pm$2 | 81$\pm$2 | 42$\pm$3 | 99$\pm$1 |
| | | Large | 0 $\pm$0 | 0 $\pm$0 | 0 $\pm$0 | 4 $\pm$5 | 21$\pm$3 | 27$\pm$1 | 13$\pm$2 | 98$\pm$1 |
| | **Average** | | 12.6 | 36.5 | 8.4 | 36.4 | 65.3 | 77.9 | 25.4 | **96.8** |

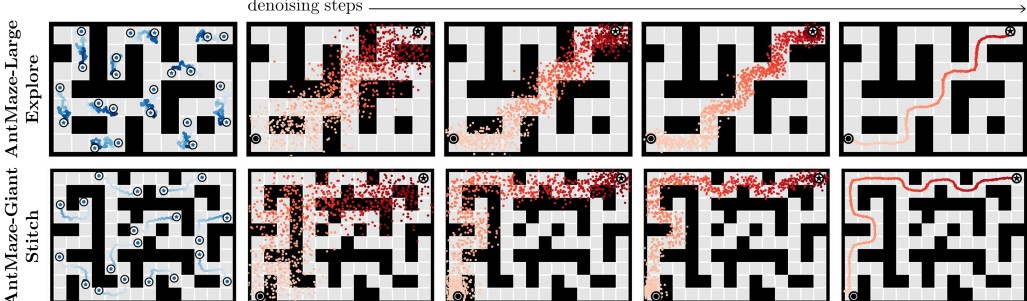

Figure 4: **SCoTS enables long-horizon planning.** We visualize trajectories generated by a diffusion planner trained on SCoTS-augmented data, evaluated on two challenging `AntMaze` datasets: `Explore` (top) and `Stitch` (bottom). The original `Stitch` dataset contains trajectories limited to four maze cells per segment, necessitating extensive stitching, whereas the `Explore` dataset comprises low-quality trajectories with large action noise. Despite these constraints, SCoTS augmentation allows the planner to synthesize trajectories that substantially surpass the horizon and quality of the original data, connecting specified start ◉ and goal ✪.

2025). The plans generated by the diffusion planner serve as sequences of subgoals for the low-level controller. At each step, the low-level controller executes actions toward a subgoal selected from the generated plan; after a fixed horizon or once the subgoal is reached, it dynamically updates the subgoal by selecting the next state at a specified horizon further along in the plan generated by the diffusion planner. For each dataset, we upsample the original data to 5M samples. Additional implementation details, including hyperparameters and specifics of the low-level controller, are provided in Appendix B.

As shown in Table 1, integrating SCoTS consistently enhances the performance of the hierarchical diffusion planner across all tasks, achieving near-optimal success rates. Notably, the advantage of SCoTS becomes especially pronounced as the complexity and scale of the mazes increase, with the gap between SCoTS and other baselines maximized in the largest (`Giant`) environments. Furthermore, in the challenging `Explore` dataset of the `AntMaze` environment consisting of noisy and short-range exploratory trajectories, augmentation via SCoTS significantly improves the planner ability to generate coherent, long-range, goal-directed plans, clearly highlighting the effectiveness of SCoTS.

## 4.4 Offline GCRL with SCoTS-Augmented Data

Although SCoTS is primarily designed for diffusion planners, we additionally evaluate whether trajectories augmented by SCoTS can enhance the performance of existing offline goal-conditioned RL (GCRL) algorithms. Specifically, we retrain widely used offline GCRL algorithms, including GCIQL (Kostrikov et al., 2022), CRL (Eysenbach et al., 2022), and HIQL (Park et al., 2023), on the SCoTS-augmented dataset. All hyperparameters remain identical to their original implementations. Additionally, we compare our approach with SynthER (Lu et al., 2023), which employs an

Table 2: **Performance enhancement of offline GCRL algorithms with SCoTS-augmented dataset.** Results are averaged over 5 seeds, each with 50 episodes per task. Standard deviations are indicated by ± sign.

| Env | Type | Size | GCIQL | | | CRL | | | HIQL | | |
|---|---|---|---|---|---|---|---|---|---|---|---|
| | | | Original | SynthER | SCoTS | Original | SynthER | SCoTS | Original | SynthER | SCoTS |
| PointMaze | Stitch | Medium | 21 ±9 | 30 ±3 | 79 ±1 | 0 ±1 | 0 ±0 | 46 ±2 | 74 ±6 | 77 ±4 | 82 ±4 |
| | | Large | 31 ±2 | 35 ±4 | 26 ±2 | 0 ±0 | 0 ±0 | 39 ±2 | 13 ±6 | 16 ±3 | 67 ±1 |
| | | Giant | 0 ±0 | 0 ±0 | 0 ±0 | 0 ±0 | 0 ±0 | 18 ±2 | 0 ±0 | 0 ±0 | 27 ±2 |
| AntMaze | Stitch | Medium | 29 ±6 | 31 ±3 | 35 ±2 | 53 ±6 | 48 ±3 | 65 ±3 | 94 ±1 | 91 ±2 | 94 ±1 |
| | | Large | 7 ±2 | 3 ±4 | 7 ±1 | 11 ±2 | 12 ±2 | 19 ±1 | 67 ±5 | 65 ±3 | 91 ±2 |
| | | Giant | 0 ±0 | 0 ±0 | 0 ±0 | 0 ±0 | 0 ±0 | 2 ±1 | 2 ±2 | 0 ±0 | 55 ±5 |
| | Explore | Medium | 13 ±2 | 12 ±3 | 18 ±3 | 3 ±2 | 3 ±1 | 15 ±3 | 37 ±10 | 45 ±8 | 94 ±1 |
| | | Large | 0 ±0 | 0 ±0 | 0 ±0 | 0 ±0 | 2 ±1 | 19 ±1 | 4 ±5 | 12 ±3 | 77 ±2 |
| | **Average** | | 12.6 | 13.9 | **20.7** | 8.4 | 8.1 | **27.9** | 36.4 | 38.3 | **73.4** |

unconditional diffusion model for transition-level data augmentation. Results summarized in Table 2 clearly demonstrate that SCoTS-generated trajectories consistently outperform SynthER and methods trained solely on the original offline datasets, significantly boosting performance across all tested algorithms. This indicates that augmenting data at the trajectory-level with SCoTS, which explicitly considers long-term dynamics and diversity, provides more effective supervision for learning robust trajectory stitching and long-horizon planning capabilities.

## 4.5 Ablation Studies

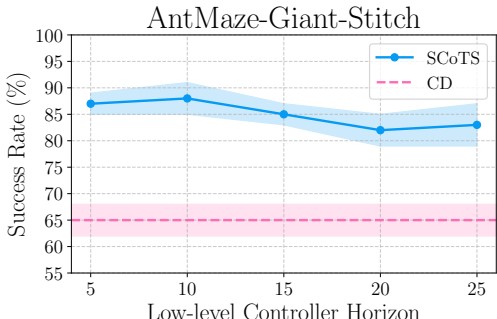

Figure 5: **Ablation study on low-level controller horizon.** Success rates in the `AntMaze-Giant-Stitch` environment comparing SCoTS against Compositional Diffuser (CD) (Luo et al., 2025), across various low-level controller horizon lengths.

Figure 6: **Dynamic MSE comparison at stitching points.** Histograms showing the distributions of Dynamic MSE at trajectory stitching points in the `AntMaze-Giant-Stitch` environment, comparing results with and without the diffusion-based stitching refinement step.

**Ablation study on low-level controller horizon.** We investigate how the performance of our approach (SCoTS) is influenced by varying the horizon length of the low-level controller in the `AntMaze-Giant-Stitch` environment. As shown in Figure 5, SCoTS achieves consistently strong performance across different horizon lengths $H \in \{5, 10, 15, 20, 25\}$, outperforming the Compositional Diffuser (CD) (Luo et al., 2025). These results demonstrate that the diffusion planner trained with SCoTS generates highly feasible subgoals, maintaining robustness and effectiveness regardless of the chosen low-level execution horizon.

**Effectiveness of diffusion-based stitching refinement.** To further illustrate the effectiveness of the diffusion-based stitching refinement step in our SCoTS framework, we quantitatively evaluate its impact on dynamic consistency at stitching points. Specifically, we compute the *Dynamic Mean Squared Error (Dynamic MSE)* (Lu et al., 2023), defined as:

$$\text{Dynamic MSE} = \|f^*(\boldsymbol{s}, \boldsymbol{a}) - \boldsymbol{s}'\|_2^2,$$

which measures how closely the generated transitions adhere to the true environment dynamics $f^*$. Figure 6 compares the distribution of Dynamic MSE at stitching points before and after applying refinement on a logarithmic scale. Results clearly show that diffusion-based refinement substantially reduces dynamic inconsistencies, highlighting its critical role in generating dynamically feasible and coherent trajectories.

**Ablation study on replanning.** We employ replanning during a rollout, enabling the agent to recover from failures, such as when the diffusion planner generates unreachable subgoals for the low-level controller. In practice, we set a replanning interval (e.g., every 200 steps); further implementation details are provided in Appendix B. In Table 3, we present an ablation study comparing performance with and without replanning on the `PointMaze` and `AntMaze Stitch` datasets from OGBench. SCoTS consistently outperforms Compositional Diffuser (CD) (Luo et al., 2025), the best-performing baseline, even without re-

Table 3: **Impact of replanning.** Success rates on OGBench `PointMaze` and `AntMaze Stitch` datasets, comparing SCoTS and CD (Luo et al., 2025). ✓ indicates with replanning; ✗ indicates without replanning.

| Env | Size | CD | | SCoTS | |
|---|---|---|---|---|---|
| | | ✗ | ✓ | ✗ | ✓ |
| PointMaze | Medium | $100_{\pm 0}$ | $100_{\pm 0}$ | $100_{\pm 0}$ | $100_{\pm 0}$ |
| | Large | $100_{\pm 0}$ | $100_{\pm 0}$ | $100_{\pm 0}$ | $100_{\pm 0}$ |
| | Giant | $53_{\pm 6}$ | $68_{\pm 3}$ | $89_{\pm 2}$ | $100_{\pm 0}$ |
| AntMaze | Medium | $92_{\pm 2}$ | $96_{\pm 2}$ | $97_{\pm 1}$ | $97_{\pm 1}$ |
| | Large | $76_{\pm 2}$ | $86_{\pm 2}$ | $92_{\pm 2}$ | $93_{\pm 1}$ |
| | Giant | $27_{\pm 4}$ | $65_{\pm 3}$ | $84_{\pm 2}$ | $87_{\pm 2}$ |
| **Average** | | 74.7 | 85.9 | 93.7 | **96.2** |

planning. Additionally, the performance with and without replanning is similar, highlighting the reliability and efficacy of the SCoTS-augmented diffusion planner.

# 5 Related Work

**Planning with Diffusion Models.** Diffusion probabilistic models (Sohl-Dickstein et al., 2015; Ho et al., 2020) have emerged as powerful tools for reinforcement learning, especially in offline settings. These models iteratively denoise sampled data from noise, effectively learning gradients of the data distribution (Song & Ermon, 2019) and demonstrating strong capabilities in modeling complex trajectories. Early work such as Diffuser (Janner et al., 2022) employed unconditional diffusion models guided by learned value estimators (Dhariwal & Nichol, 2021). Subsequent methods like Decision Diffuser (Ajay et al., 2023) and AdaptDiffuser (Liang et al., 2023) introduced classifier-free guidance and progressive fine-tuning. Recent advancements further leveraged hierarchical structures (Chen et al., 2024c; Li et al., 2023), multi-task conditioning (Ni et al., 2023; He et al., 2023; Dong et al., 2024), and multi-agent setups (Zhu et al., 2023). Additionally, diffusion planners have explored integration with tree search methods (Yoon et al., 2025a,b), refined trajectory sampling techniques (Lee et al., 2023b; Feng et al., 2024; Lee & Choi, 2025), and investigated critical design choices to improve robustness (Lu et al., 2025). Despite these advances, diffusion planners still fundamentally depend on the quality and diversity of the offline training datasets, limiting their ability to generate coherent and feasible long-horizon plans beyond their training distribution. Recent approaches such as Generative Skill Chaining (GSC) (Mishra et al., 2023) and Compositional Diffuser (Luo et al., 2025) address this by composing short segments at test time into long-horizon trajectories. Our work presents an orthogonal solution by directly augmenting the offline dataset itself, significantly enhancing the capability of diffusion planners to generalize to diverse and substantially longer trajectories.

**Data Augmentation for RL.** Data augmentation is a recognized strategy for improving sample efficiency and generalization in reinforcement learning (RL). In pixel-based RL, techniques like random image transformations (e.g., cropping, translation) have proven effective in works such as CURL (Laskin et al., 2020b), RAD (Laskin et al., 2020a), and DrQ (Yarats et al., 2021). For state-based observations, methods like S4RL (Sinha et al., 2022) and AWM (Ball et al., 2021) often introduce perturbations to states or learned dynamics models to enhance robustness. Recent advances in generative models have enabled trajectory-level augmentation methods, either at the transition level (Lu et al., 2023; Wang et al., 2024) or the full trajectory level (He et al., 2023; Jackson et al., 2024; Lee et al., 2024). For instance, MTDiff-S (He et al., 2023) generates synthetic trajectories for multi-task scenarios, while Policy-Guided Diffusion (PGD) (Jackson et al., 2024) and GTA (Lee et al., 2024) employ generative models to produce high-reward trajectories guided by policies or returns.

DiffStitch (Li et al., 2024) further systematically connects trajectories based on extrinsic rewards, yet these methods typically require explicit reward signals and are limited to generating short-horizon trajectories. In contrast, our proposed SCoTS method operates in a *reward-free* manner, systematically synthesizing long-horizon, diverse, and dynamically consistent trajectories to significantly enhance offline datasets, thereby facilitating the generation of feasible plans in downstream tasks requiring extended horizon reasoning.

**Temporal Distance in RL.** Temporal distance has been widely adopted as a structural inductive bias in various reinforcement learning (RL) paradigms, including imitation learning (Sermanet et al., 2018), unsupervised skill discovery (Hartikainen et al., 2019; Park et al., 2024b,a), goal-conditioned RL (Durugkar et al., 2021; Eysenbach et al., 2022; Wang et al., 2023; Bae et al., 2024), and curriculum learning (Zhang et al., 2020; Kim et al., 2023). Recent methods such as METRA (Park et al., 2024b), QRL (Wang et al., 2023), HILP (Park et al., 2024a), and TLDR (Bae et al., 2024) particularly focus on learning temporal distance-preserving representations to facilitate diverse skill discovery or efficient goal-reaching behaviors. Distinct from prior methods, our SCoTS framework explicitly leverages temporal distance-preserving representations to identify temporally viable trajectory segments for stitching. This allows systematic synthesis of extended, diverse, and dynamically consistent trajectories, significantly augmenting offline datasets and improving long-horizon generalization for diffusion-based planners.

# 6   Conclusion

In this work, we introduced *State-Covering Trajectory Stitching* (SCoTS), a novel reward-free trajectory augmentation approach designed to enhance the performance and generalization capabilities of diffusion planners. By leveraging temporal distance-preserving embeddings, SCoTS iteratively stitches together short trajectory segments, systematically extending the diversity and horizon of offline data. Empirical results across challenging benchmarks demonstrated that SCoTS-generated trajectories significantly improve the ability of diffusion planners to perform long-horizon planning and generalize to novel tasks. Furthermore, we showed that our augmented dataset notably enhances the performance of widely used offline goal-conditioned reinforcement learning algorithms, highlighting the broad utility of our approach.

**Limitations.** While SCoTS achieves strong empirical performance, it exhibits certain limitations. First, generating augmented trajectories through iterative stitching and diffusion-based refinement introduces significant computational overhead, especially due to the additional training of the diffusion-based stitcher model and the trajectory augmentation process. Second, our temporal distance-preserving embeddings do not capture the asymmetric temporal distances between states, potentially limiting their effectiveness in highly asymmetric or disconnected Markov Decision Processes (MDPs), such as object manipulation tasks involving irreversible actions or environments containing isolated regions with sparse connectivity.

**Impact Statement.** This paper advances the field of diffusion-based planning by introducing a novel trajectory augmentation method, enhancing long-horizon reasoning capabilities in reinforcement learning. While we do not identify direct negative societal impacts stemming from this research, practitioners are encouraged to carefully assess real-world implications, particularly regarding safety and reliability, prior to deploying this method in practical scenarios.

# Acknowledgements

This work was supported by Institute of Information & communications Technology Planning & Evaluation (IITP) grant funded by the Korea government (MSIT) (No.2019-0-00075, Artificial Intelligence Graduate School Program (KAIST); No. 2022-0-00984, Development of Artificial Intelligence Technology for Personalized Plug-and-Play Explanation and Verification of Explanation; No. RS-2024-00457882, AI Research Hub Project; No.RS-2022-II220184, Development and Study of AI Technologies to Inexpensively Conform to Evolving Policy on Ethics), and by the InnoCORE program of the Ministry of Science and ICT (N10250156).

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

# A Details of Datasets

We evaluate our method on the OGBench benchmark (Park et al., 2025)[1]. Since our primary goal is to assess trajectory stitching capability and long-horizon reasoning, we specifically utilize the `Stitch` and `Explore` datasets. As shown in Figure 7, the `Stitch` dataset is explicitly designed to challenge trajectory stitching ability, comprising short, goal-reaching trajectories limited to a maxi-

Table 4: **Dataset specifications.**

| Env | Type | Size | # Transitions | # Episodes | Data Episode Length |
|---|---|---|---|---|---|
| PointMaze | Stitch | Medium | 1M | 5,000 | 200 |
| | | Large | 1M | 5,000 | 200 |
| | | Giant | 1M | 5,000 | 200 |
| AntMaze | Stitch | Medium | 1M | 5,000 | 200 |
| | | Large | 1M | 5,000 | 200 |
| | | Giant | 1M | 5,000 | 200 |
| | Explore | Medium | 5M | 10,000 | 500 |
| | | Large | 5M | 10,000 | 500 |

mum length of four cell units. Consequently, agents must effectively stitch together multiple short segments (up to eight) to successfully complete long-horizon tasks. In contrast, the `Explore` dataset is designed to test navigation skills learned from extensive yet low-quality trajectories. These trajectories are generated by commanding a low-level policy with random movement directions re-sampled every ten steps, along with significant action noise. Each demonstration trajectory typically spans only two to three blocks, resulting in noisy and clustered paths that pose additional challenges for evaluating the ability to learn effective policies from highly suboptimal data.

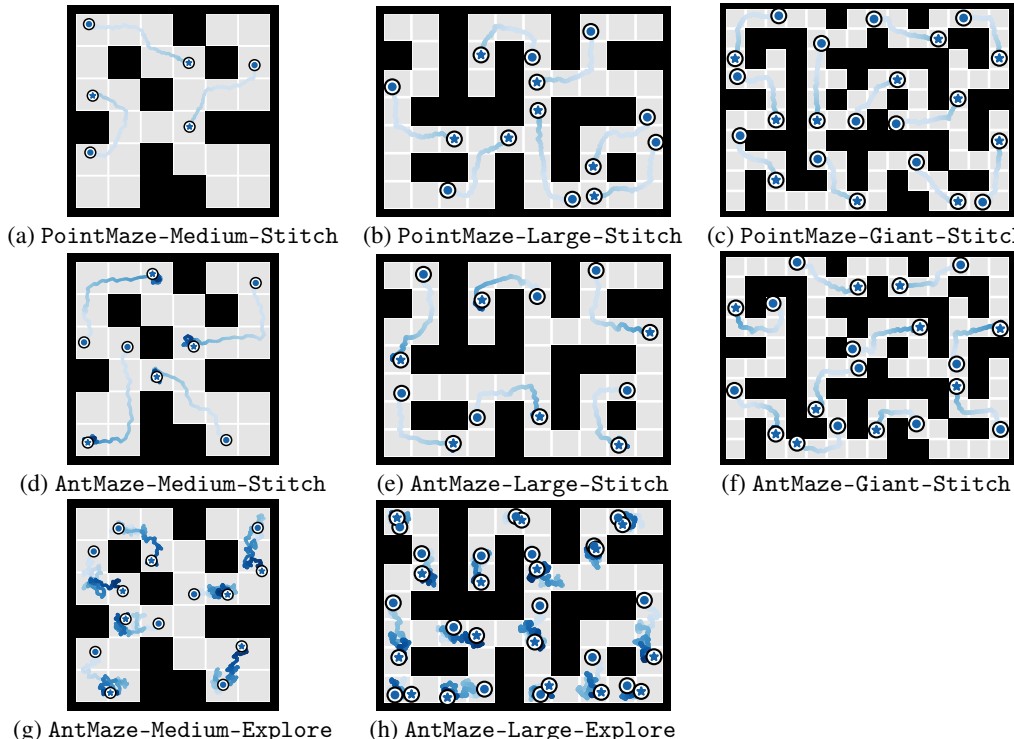

(a) PointMaze-Medium-Stitch    (b) PointMaze-Large-Stitch    (c) PointMaze-Giant-Stitch

(d) AntMaze-Medium-Stitch    (e) AntMaze-Large-Stitch    (f) AntMaze-Giant-Stitch

(g) AntMaze-Medium-Explore    (h) AntMaze-Large-Explore

Figure 7: **Visualization of trajectories from OGBench datasets.** Each sub-figure illustrates example trajectories from different combinations of environments and datasets used in our experiments.

# B Implementation Details

**Network architecture.** We utilize DiT1D (Peebles & Xie, 2023) as the neural network backbone for both the diffusion planner and the stitcher, due to its large receptive field and effectiveness in modeling trajectory-level dependencies. Following prior studies (Dong et al., 2023; Lu et al., 2025), we employ a DiT1D architecture with a hidden dimension of 256, a head dimension of 32, and a total of 8 DiT blocks consistently across all environments.

---

[1]https://github.com/seohongpark/ogbench

Table 5: **Hyperparameters for SCoTS.**

| Component | Hyperparameter | Value | Tuning Choices |
|---|---|---|---|
| *SCoTS: Temporal Distance-Preserving Embedding ($\phi$)* | | | |
| | Learning Rate | $3 \times 10^{-4}$ | - |
| | Latent Dimension | 32 | - |
| | Batch Size | 1024 | - |
| | Training Steps | 1,000,000 | - |
| | Network Backbone | MLP | - |
| | MLP Dimensions | (512, 512, 512) | - |
| | Expectile ($\xi$ for $\ell_\xi^2$) | 0.95 | - |
| *SCoTS: Inverse Dynamics Model (for actions in $\mathcal{D}_{aug}$)* | | | |
| | Network Backbone | MLP | - |
| | MLP Dimensions | (256, 256, 256) | - |
| | Training Steps | 200,000 | - |
| *SCoTS: Stitching Process Parameters* | | | |
| | Top-$k$ Candidates (Search) | 10 | - |
| | $k_{\text{density}}$ (Novelty Score) | 30 | - |
| | Novelty Weight ($\beta$) | 2.0 | - |
| | Augmented Dataset Size | $\sim$5M transitions | - |
| | $N_{\text{stitch}}$ (Stitches per Traj.) | Task-dependent (e.g., 40) | - |
| | $N_{\text{traj}}$ (Generated Traj.) | Task-dependent (e.g., 5,000) | - |
| *SCoTS: Diffusion-based Stitcher ($p_\theta^{stitcher}$)* | | | |
| | Network Backbone | DiT1D | - |
| | Learning Rate | $2 \times 10^{-4}$ | - |
| | Weight Decay | $1 \times 10^{-5}$ | - |
| | Batch Size | 64 | - |
| | Training Steps | 1,000,000 | - |
| | Solver | DDIM | - |
| | Sampling Steps (DDIM) | 20 | - |
| | Horizon ($H_{\text{stitcher}}$) | 26 | - |
| *Hierarchical Diffusion Planner (HD)* | | | |
| | Network Backbone | DiT1D | - |
| | Learning Rate | $2 \times 10^{-4}$ | - |
| | Weight Decay | $1 \times 10^{-5}$ | - |
| | Batch Size | 64 | - |
| | Training Steps | 1,000,000 | - |
| | Solver | DDIM | - |
| | Sampling Steps (DDIM) | 20 | - |
| | Plan Horizon (on original data) | 101 (`Stitch`), 401 (`Explore`) | - |
| | Plan Horizon (on $\mathcal{D}_{\text{aug}}$) | 501 (M/L), 1001 (G/Explore) | - |
| | Temporal Jump | 26 | - |
| *Execution Parameters* | | | |
| | Low-level Controller Horizon | Tuned | $\{5, 10, 15, 20, 25\}$ |
| | Replanning Interval | Tuned | $\{50, 100, 200\}$ |

**Details of the low-level controller.** A key challenge in diffusion-based planning is balancing global trajectory coherence with effective low-level control in high-dimensional state-action spaces (Chen et al., 2024a,b; Yoon et al., 2025a). Previous approaches, such as PlanDQ (Chen et al., 2024b) and MCTD (Yoon et al., 2025a), address this issue by integrating high-level diffusion planners with separately trained low-level controllers. Similarly, we adopt a hierarchical strategy, where the diffusion planner generates plans based primarily on compact, lower-dimensional state representations (e.g., positions of the agent itself), delegating the fine-grained, low-level action execution to a dedicated low-level controller. In our experiments, we specifically employ GCIQL (Kostrikov et al., 2022) as the learned low-level policy in the `PointMaze` environments and CRL (Eysenbach et al., 2022) in the `AntMaze` environments. A detailed visualization of generated subgoals and their corresponding execution rollouts can be seen in Figure 9. Furthermore, an ablation study examining the impact of the horizon length of the low-level controller is presented in Figure 5.

**Implementation details for SCoTS and diffusion planning.** In the temporal distance-preserving search stage of SCoTS, we retrieve the top $k = 10$ candidate segments based on their proximity in the learned latent embedding space during each stitching step. For computing the novelty score, we utilize a density estimator parameter $k_{\text{density}} = 30$ and set the novelty weighting factor $\beta = 2.0$ consistently across all tested environments. The horizon length for the diffusion-based stitcher is uniformly set to $H_{\text{stitcher}} = 26$.

To generate the augmented dataset $\mathcal{D}\text{aug}$, we perform the stitching procedure $N\text{stitch}$ iterations per trajectory, creating a total of $N_{\text{traj}}$ trajectories, thus ensuring the augmented dataset comprises approximately 5 million transitions. Specifically, in the `AntMaze-Large-Stitch` environment, we set $N_{\text{stitch}} = 40$ and $N_{\text{traj}} = 5000$.

For configuring the Hierarchical Diffusion (HD) planner (Chen et al., 2024c), parameters are adapted according to the properties of the training data. When training on the original `Stitch` and `Explore` datasets, which contain inherently shorter trajectories (as detailed in Table 4, column "Data Episode Length"), we set the high-level planning horizon to 101 steps for `Stitch` and 401 steps for `Explore`, both with temporal jumps of 26 steps between waypoints. However, when utilizing SCoTS-augmented datasets that feature longer and more diverse trajectories, we extend this planning horizon to 501 steps for `Medium` and `Large` environments, and to 1001 steps for `Giant` environments, maintaining the temporal jump of 26 steps. Similarly, for SCoTS-augmented `Explore` datasets, we also use a planning horizon of 1001 steps with 26-step jumps.

We apply jumpy denoising with DDIM sampling (Song et al., 2020) using 20 denoising steps across all environments. Additionally, we tune the replanning interval from the set $\{50, 100, 200\}$ steps and tune the horizon for the low-level controller from $\{5, 10, 15, 20, 25\}$. A full list of the hyperparameters is reported in Table 5.

**Practical implementation of temporal distance-preserving search.** Our SCoTS framework relies on a learned latent space $\mathcal{Z}$ where the $L_2$ distance, $\|\phi(\boldsymbol{s}) - \phi(\boldsymbol{g})\|_2$, approximates the optimal temporal distance $d^*(\boldsymbol{s}, \boldsymbol{g})$ between states (as detailed in Section 3.1). A critical step in SCoTS is the efficient identification of suitable candidate trajectory segments from a large offline dataset $\mathcal{D}$. This requires a fast nearest neighbor search mechanism within the learned latent space $\mathcal{Z}$. To achieve this, we employ an Inverted File (IVF) index from the Faiss library (Douze et al., 2024), which is specifically designed for large-scale similarity searches.

The practical implementation of this search mechanism involves several stages. First, we prepare the data for indexing. This consists of computing the latent embeddings $\phi(\boldsymbol{s}_{\text{init}})$ for the initial states $\boldsymbol{s}_{\text{init}}$ of all trajectories within the offline dataset $\mathcal{D}$. Let $d$ denote the dimensionality of these latent embeddings. An IVF index is then constructed upon this collection of $d$-dimensional vectors. The construction process begins by partitioning the latent vectors into $n_{\text{list}}$ clusters using the $k$-means algorithm. Each cluster is represented by a centroid $\mathbf{c}_j \in \{\mathbf{c}_1, \ldots, \mathbf{c}_{n_{\text{list}}}\}$. Subsequently, each latent vector $\phi(\boldsymbol{s}_{\text{init}})$ in our collection is assigned to its nearest centroid, and for each centroid, an inverted list is maintained, storing references to the vectors belonging to its cluster.

During the temporal distance-preserving search phase of SCoTS (detailed in Algorithm 1, line 10), the latent embedding of the current composed trajectory endpoint, $\phi(\text{end}(\boldsymbol{\tau}_{\text{comp}}))$, serves as the query vector $\mathbf{q}$. To find the $k$ nearest neighbors for $\mathbf{q}$, the IVF index first identifies a limited set of clusters whose centroids $\{\mathbf{c}_j\}$ are closest to the query vector $\mathbf{q}$. The search for neighbors is then confined to

the latent vectors stored within the inverted lists corresponding to these selected clusters. This targeted approach significantly prunes the search space compared to an exhaustive search. Furthermore, the Faiss library provides support for GPU acceleration, which can further expedite this search process and enable efficient candidate retrieval. Once the $k$ nearest latent embeddings corresponding to initial states of segments are identified, we retrieve the full original trajectory segments from $\mathcal{D}$ to form the candidate set for the stitching process.

**Computational resources and runtimes.** All experiments were conducted using a single NVIDIA A10 GPU. The approximate execution times for each component of our method are as follows:

- Temporal distance-preserving embedding training: 1.5 hours
- Inverse dynamics model training: 0.25 hours
- Low-level controller training: 2.5 hours
- Diffusion-based stitcher training: 7 hours
- Trajectory augmentation via SCoTS: 0.5 hours
- Diffusion planner training: 18 hours

These times are per model training instance or data generation run and may vary slightly depending on the specific environment and dataset characteristics.

## C  Additional Results

**Ablation study on temporal distance-preserving embedding.** To isolate the contribution of our learned latent space, we compare the performance of the HD (Chen et al., 2024c) planner trained on SCoTS-augmented data where the stitching process was guided by either our temporal distance-preserving embedding or raw state space. As shown in Table 6, temporal distance-preserving representation is crucial for effective stitching. This is especially true in high-dimensional environments like `AntMaze`, where a small Euclidean distance in the raw state space (e.g., between two similar joint configurations) does not guarantee reachability. Even in `Pointmaze`, where state-space distance is more intuitive, the learned compact latent space provides a better-structured representation for stitching temporally extended trajectories.

Table 6: **Ablation on the temporal distance-preserving embedding.** Success rates on OGBench `PointMaze` and `AntMaze Stitch` datasets. ✗ indicates stitching guided by raw state-space distances (w/o temporal embedding), while ✓ indicates guidance from learned temporal embedding.

| Env | Size | SCoTS ✗ | SCoTS ✓ |
|---|---|---|---|
| PointMaze | Large | $93_{\pm 0}$ | $100_{\pm 0}$ |
| | Giant | $52_{\pm 1}$ | $100_{\pm 0}$ |
| AntMaze | Large | $45_{\pm 1}$ | $93_{\pm 1}$ |
| | Giant | $7_{\pm 2}$ | $87_{\pm 2}$ |
| **Average** | | 49.3 | **95.0** |

**Ablation study on diffusion-based stitching refinement.** To demonstrate the importance of the refinement step, we compare the performance of both a diffusion planner (HD) and a GCRL algorithm (HIQL) trained on SCoTS data generated with and without the diffusion-based refinement. The results, summarized in Table 7, reveal the critical importance of this component. Without refinement, the connection points between stitched segments can suffer from large dynamic inconsistency errors, as illustrated in Figure 6. This performance degradation is particularly significant when training GCRL algorithms like HIQL, which are highly sensitive to the dynamic validity of the training transitions.

Table 7: **Ablation on diffusion-based stitching refinement.** Success rates on `AntMaze Stitch` datasets for HD (Chen et al., 2024c) and HIQL (Park et al., 2023). ✗ indicates training on SCoTS data generated without refinement, while ✓ indicates training on data generated with refinement.

| Env | Size | HD ✗ | HD ✓ | HIQL ✗ | HIQL ✓ |
|---|---|---|---|---|---|
| AntMaze | Large | $85_{\pm 3}$ | $93_{\pm 1}$ | $52_{\pm 2}$ | $91_{\pm 2}$ |
| | Giant | $53_{\pm 1}$ | $87_{\pm 2}$ | $11_{\pm 2}$ | $55_{\pm 5}$ |
| **Average** | | 69.0 | **90.0** | 31.5 | **73.0** |

**Sensitivity analysis on the novelty weight $\beta$.** As illustrated in Figure 3, using only the progress score ($\beta = 0$), which quantifies the alignment with a latent exploration direction, is effective at generating long, temporally-extended trajectories with clear directional distinctions. However, relying solely on this directional guidance can be suboptimal. The learned temporal distance-preserving embedding $\phi$ is a powerful but imperfect approximation of the true environment topology. Due to embedding errors, the geometry of the latent space can be distorted. Consequently, a straight line between two latent states, $\phi(s)$ and $\phi(s')$, may not correspond to a feasible path in the underlying MDP. Therefore, exploratory detours are often necessary to find a valid temporally extended path. This is precisely where the novelty score becomes critical. It encourages these exploratory detours by rewarding the selection of segments that lead to less-visited states. This allows the agent to navigate around the imperfections and distortions in the learned latent space, discovering feasible and often more effective paths. To demonstrate this complementary effect quantitatively, we conducted an additional ablation study on 2 OGBench tasks. The table below compares the performance of the HD planner trained with SCoTS data generated using different novelty weights $\beta$ (5 seeds). The results, summarized in Table 8, suggest that while directional stitching alone ($\beta = 0$) provides a notable performance improvement, introducing and balancing it with the novelty score ($\beta > 0$) yields substantial further gains. This effect is especially pronounced on the more complex `Giant` task.

Table 8: **Sensitivity to novelty weight $\beta$.**

| **Env** | **Type** | **Size** | **HD** | $\beta$=0 | $\beta$=2 | $\beta$=4 | $\beta$=8 |
|---------|----------|----------|--------|-----------|-----------|-----------|-----------|
| AntMaze | Stitch | Large | $36_{\pm2}$ | $87_{\pm2}$ | $93_{\pm1}$ | $92_{\pm1}$ | $85_{\pm3}$ |
|         |        | Giant | $0_{\pm0}$ | $63_{\pm3}$ | $87_{\pm2}$ | $89_{\pm2}$ | $74_{\pm3}$ |

**Sensitivity analysis on sub-trajectory length $H$.** As shown in Table 9, performance on the `Stitch` dataset is relatively robust to the choice of $H$. However, on the `Explore` dataset, shorter segments perform best. We hypothesize this is because the `Explore` dataset consists of low-quality, noisy trajectories. Using shorter segments allows SCoTS to be more selective, finding and connecting the temporally-extended parts of trajectories.

Table 9: **Sensitivity to sub-trajectory length $H$.**

| **Env** | **Type** | **Size** | $H$=26 | $H$=52 | $H$=104 |
|---------|----------|----------|--------|--------|---------|
| AntMaze | Stitch | Giant | $87_{\pm2}$ | $89_{\pm2}$ | $87_{\pm2}$ |
|         | Explore | Large | $98_{\pm1}$ | $93_{\pm1}$ | $88_{\pm2}$ |

**Sensitivity analysis on the number of retrieved segments $K$.** Table 10 shows that performance is robust as long as $K$ is not too small. A very small $K$ limits the diversity of candidate segments, hindering the effectiveness of our progress and novelty-based selection. While performance is stable for larger $K$, we expect an excessively large $K$ could eventually degrade performance by increasing the chance of retrieving dynamically inconsistent segments.

Table 10: **Sensitivity to number of retrieved segments $K$.**

| **Env** | **Type** | **Size** | $K$=3 | $K$=10 | $K$=20 |
|---------|----------|----------|-------|--------|--------|
| AntMaze | Stitch | Giant | $65_{\pm3}$ | $87_{\pm2}$ | $89_{\pm2}$ |
|         | Explore | Large | $72_{\pm2}$ | $98_{\pm1}$ | $97_{\pm1}$ |

**Evaluation on dense-reward offline RL benchmarks.** While our paper focuses on tasks where extrinsic rewards are absent, the SCoTS framework is not limited to goal-conditioned tasks. To demonstrate its applicability to general, dense-reward offline RL, we conducted additional experiments on four standard MuJoCo locomotion benchmarks from D4RL (Kumar et al., 2020), utilizing suboptimal `medium` and `medium-replay` datasets. It is important to clarify that our use of *reward-free* refers to the stitching mechanism itself, which is guided by the intrinsic temporal structure of the data rather than extrinsic task rewards. This allows the core data augmentation to be task-agnostic.

For a fair comparison, we used the same Diffuser architecture and planning horizon (32) as described in the original paper (Janner et al., 2022). We upsample the original data to 2M samples using the same SCoTS procedure, with sub-trajectory lengths set to 4. Additionally, to label these new trajectories with rewards, we additionally trained a reward model, $r_t = f_\omega(s_t, a_t)$, on the original offline dataset. As shown in Table 11, training on SCoTS-augmented data consistently improves the performance of the original Diffuser. Even without architectural or sampling strategy changes,

by stitching sub-trajectories in a way that is temporally extended and exploratory, SCoTS generates novel behaviors (e.g., trajectories that travel further) that are not present in the original suboptimal dataset.

Table 11: **Performance on D4RL benchmarks.** We report normalized average returns of the Diffuser and Diffuser trained on SCoTS-augmented data. Results are averaged over 50 planning seeds.

| Dataset | Diffuser | Diffuser + SCoTS (ours) |
|---|---|---|
| `hopper-medium-v2` | $74.3_{\pm 1.4}$ | $93.5_{\pm 1.2}$ |
| `walker2d-medium-v2` | $79.6_{\pm 0.6}$ | $80.2_{\pm 0.5}$ |
| `hopper-medium-replay-v2` | $93.6_{\pm 0.4}$ | $96.7_{\pm 0.6}$ |
| `walker2d-medium-replay-v2` | $70.6_{\pm 1.6}$ | $78.5_{\pm 1.9}$ |
| **Average** | **79.5** | **87.2** |

**Evaluation on noisy and partially observable navigation.** To evaluate the robustness of SCoTS on noisy and low-quality data beyond the controlled OGBench settings, we conducted additional experiments using a mobile robot navigation task in realistic, large-scale environments (a 35m x 39m corridor and a 50m x 110m building), following the setup from HRL (Lee et al., 2023a). We simulated low-quality data by generating short, disconnected segments. Specifically, trajectories were collected by randomly selecting start and goal locations within short ranges (5m) using HRL (Lee et al., 2023a). To handle the noisy raw 2D LiDAR measurements (512 rays over 360-degree), we converted these scans into 64x64 occupancy grid images. These images were then encoded using a pre-trained $\beta$-VAE (Higgins et al., 2017) into a compact 8-dimensional latent representation, providing structured, low-dimensional state inputs for the agent. Subsequently, the SCoTS stitching method was applied to augment this fragmented dataset. The results, shown in Table 12, clearly indicate that SCoTS effectively enhances performance, even with highly fragmented and noisy datasets, demonstrating its robustness and practical applicability to robot navigation tasks.

Table 12: **Performance on a mobile robot navigation task.** Goal-reaching success rates are reported over 100 episodes for the HIQL (Park et al., 2023) trained with and without SCoTS augmentation.

| Environment | HIQL | HIQL + SCoTS (ours) |
|---|---|---|
| `Building (50m × 110m)` | $27_{\pm 7}$ | $53_{\pm 4}$ |
| `Corridor (35m × 39m)` | $0_{\pm 0}$ | $31_{\pm 5}$ |

**Visualization of temporal distance-preserving latent representations.** We train temporal distance-preserving latent representations with dimension 32 across all environments. To visualize these learned representations, we apply a $t$-distributed stochastic neighbor embedding (t-SNE) to project the 32-dimensional latent vectors onto a 2-dimensional plane, as shown in Figure 8. Recall from Equation 6 that we parameterize a goal-conditioned value function $V(\boldsymbol{s}, \boldsymbol{g})$ following (Park et al., 2024a):

$$V(\boldsymbol{s}, \boldsymbol{g}) := -||\phi(\boldsymbol{s}) - \phi(\boldsymbol{g})||_2, \tag{13}$$

which approximates the optimal goal-conditioned value function, defined as the maximum possible return (cumulative sum of rewards) for sparse-reward settings. Specifically, an agent receives a reward of 0 if the $l_2$ distance between states $\boldsymbol{s}$ and $\boldsymbol{g}$ is within a small threshold $\delta_g$, and $-1$ otherwise. The embedding function $\phi$ is trained using a temporal-difference objective inspired by implicit Q-learning (Kostrikov et al., 2022) on the offline dataset $\mathcal{D}$. As illustrated in Figure 8, the learned representations effectively capture the temporal proximity between states, resulting in latent spaces where states that are temporally close in the environment are also clustered closely in the embedding space.

**Visualization of rollout execution.** We visualize a generated plan by the diffusion planner trained on SCoTS-augmented data, along with its corresponding rollout execution in the `AntMaze-Giant-Stitch` environment, as illustrated in Figure 9. The initial image (top-left) shows the overall planned trajectory generated by the diffusion planner, with subgoals marked by green spheres. Subsequent images provide sequential snapshots from the rollout execution, demonstrating the agent actively pursuing and reaching these subgoals. This visualization highlights how effectively the generated high-level plan guides the low-level controller during task execution.

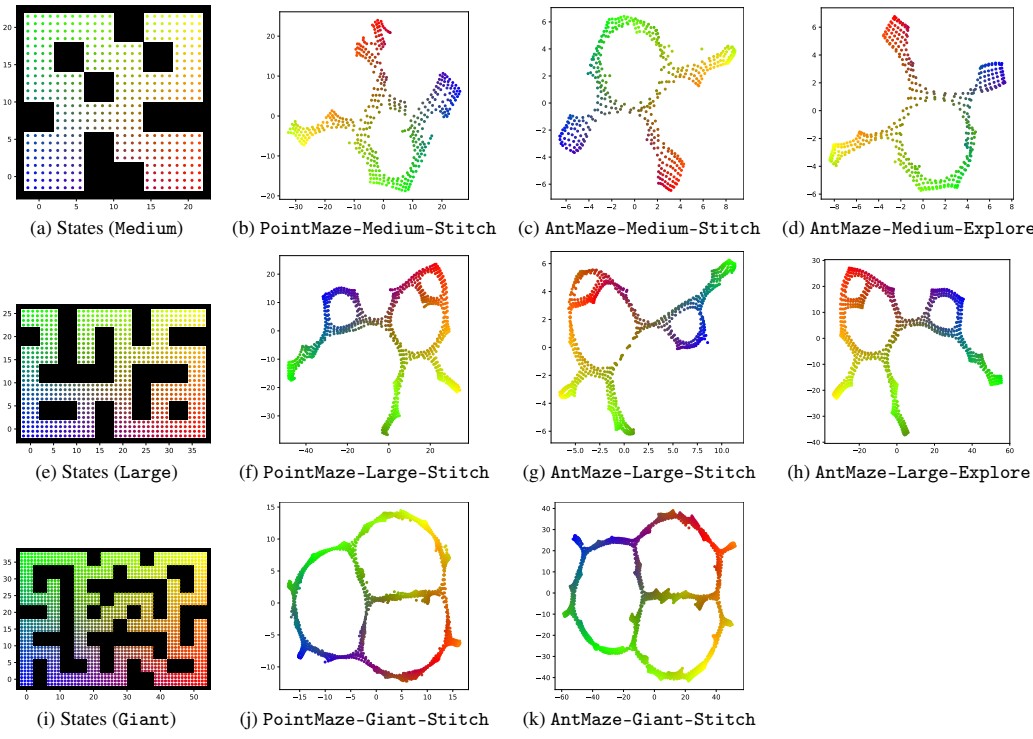

(a) States (Medium)    (b) PointMaze-Medium-Stitch    (c) AntMaze-Medium-Stitch    (d) AntMaze-Medium-Explore

(e) States (Large)    (f) PointMaze-Large-Stitch    (g) AntMaze-Large-Stitch    (h) AntMaze-Large-Explore

(i) States (Giant)    (j) PointMaze-Giant-Stitch    (k) AntMaze-Giant-Stitch

Figure 8: **Visualization of learned temporal distance-preserving latent representations.** The left-most column shows original states from maze environments of varying sizes (Medium, Large, Giant). Subsequent columns illustrate t-SNE projections of latent embeddings $\phi(s)$ for corresponding OG-Bench datasets, maintaining the same color scheme for consistency. This visualization demonstrates how spatial proximity and structure in the original state space are preserved and reflected in the learned latent representations.

**Visualization of trajectories generated by SCoTS.** In Figure 10, 11, and 12, we present representative examples of trajectories synthesized by our SCoTS framework across all considered environments and dataset types. Compared to the original trajectories provided in Figure 7, the SCoTS-generated trajectories clearly demonstrate extended coverage, illustrating the effectiveness of our method in augmenting the original offline datasets.

# D   Baseline Performance Sources

Performance scores reported for offline goal-conditioned reinforcement learning (GCRL) methods, including Goal-Conditioned Implicit Q-Learning (GCIQL) (Kostrikov et al., 2022), Quasimetric RL (QRL) (Wang et al., 2023), Contrastive RL (CRL) (Eysenbach et al., 2022), and Hierarchical Implicit Q-Learning (HIQL) (Park et al., 2023), are sourced from Table 2 in Park et al. (2025). Scores for diffusion-based generative planning methods explicitly designed for long-horizon generalization, including Generative Skill Chaining (GSC) (Mishra et al., 2023) and Compositional Diffuser (CD) (Luo et al., 2025), are sourced from Tables 1 and 2 in Luo et al. (2025).

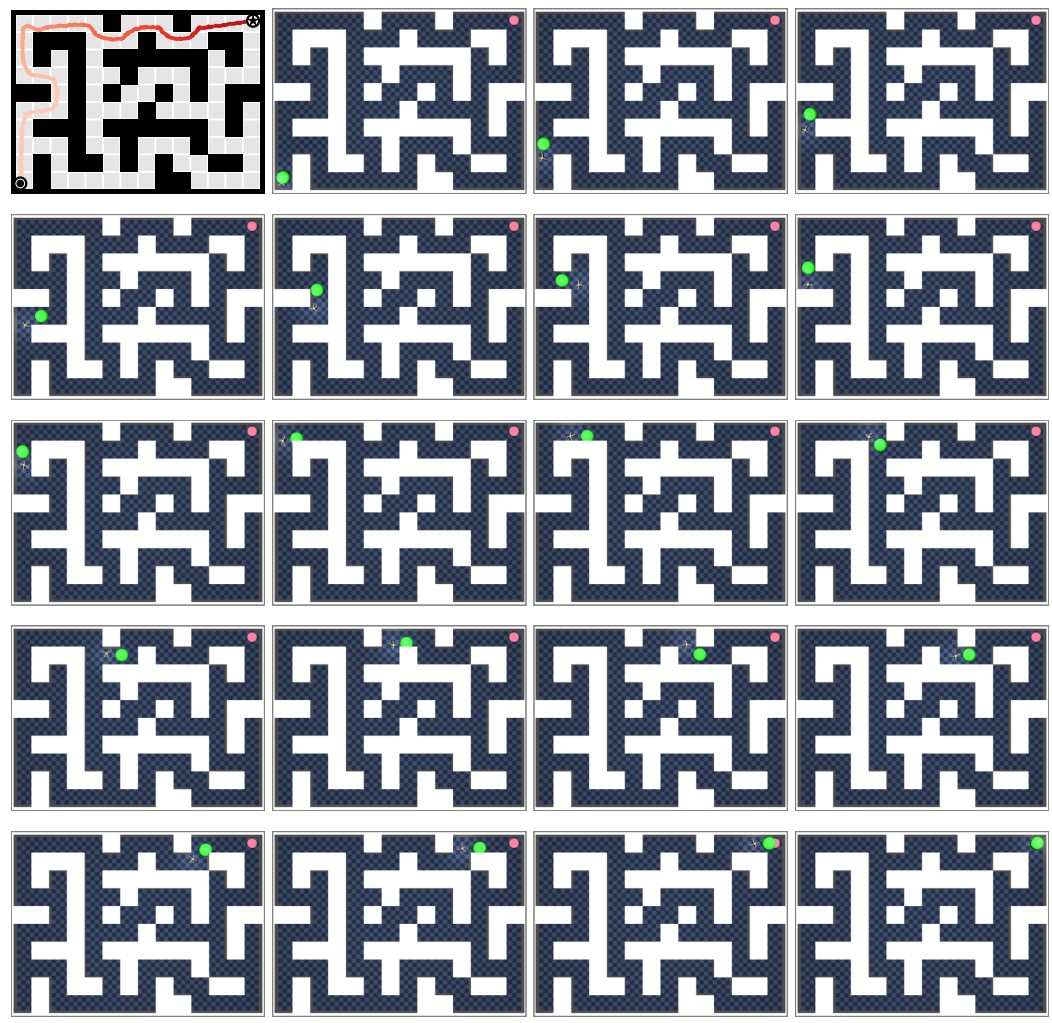

Figure 9: **Visualization of diffusion Planner rollout execution.** The top left image shows the planned trajectory generated by the diffusion planner, with subgoals marked by green spheres. Subsequent images sequentially illustrate the agent progressing toward these subgoals in the `AntMaze-Giant-Stitch` environment, demonstrating effective guidance provided by the generated plan.

Figure 10: **SCoTS-augmented trajectories for PointMaze Stitch datasets.** For each PointMaze Stitch dataset, the leftmost column shows trajectories from the original OGBench dataset. The subsequent columns are examples of SCoTS-generated trajectories.

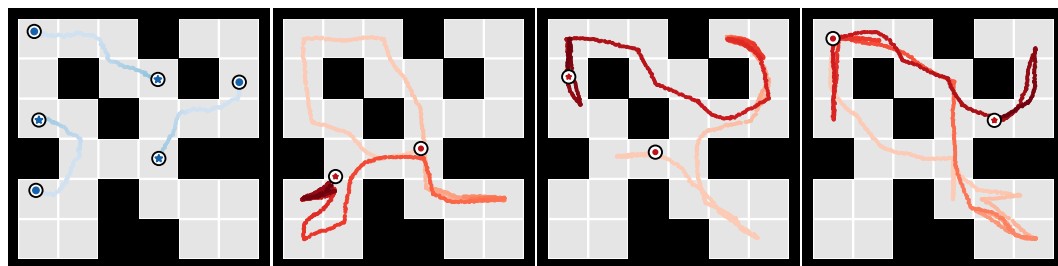

**(a)** `PointMaze-Medium-Stitch`

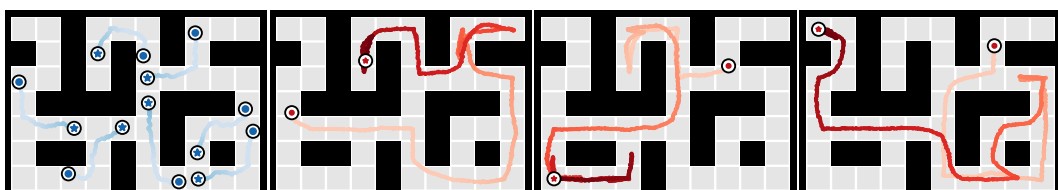

**(b)** `PointMaze-Large-Stitch`

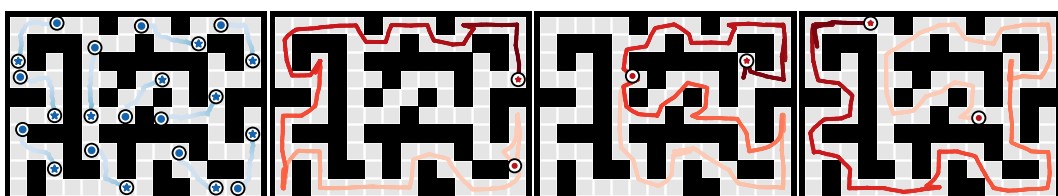

**(c)** `PointMaze-Giant-Stitch`

Figure 11: **SCoTS-augmented trajectories for AntMaze Stitch datasets.** For each AntMaze Stitch dataset, the leftmost column shows trajectories from the original OGBench dataset. The subsequent columns are examples of SCoTS-generated trajectories.

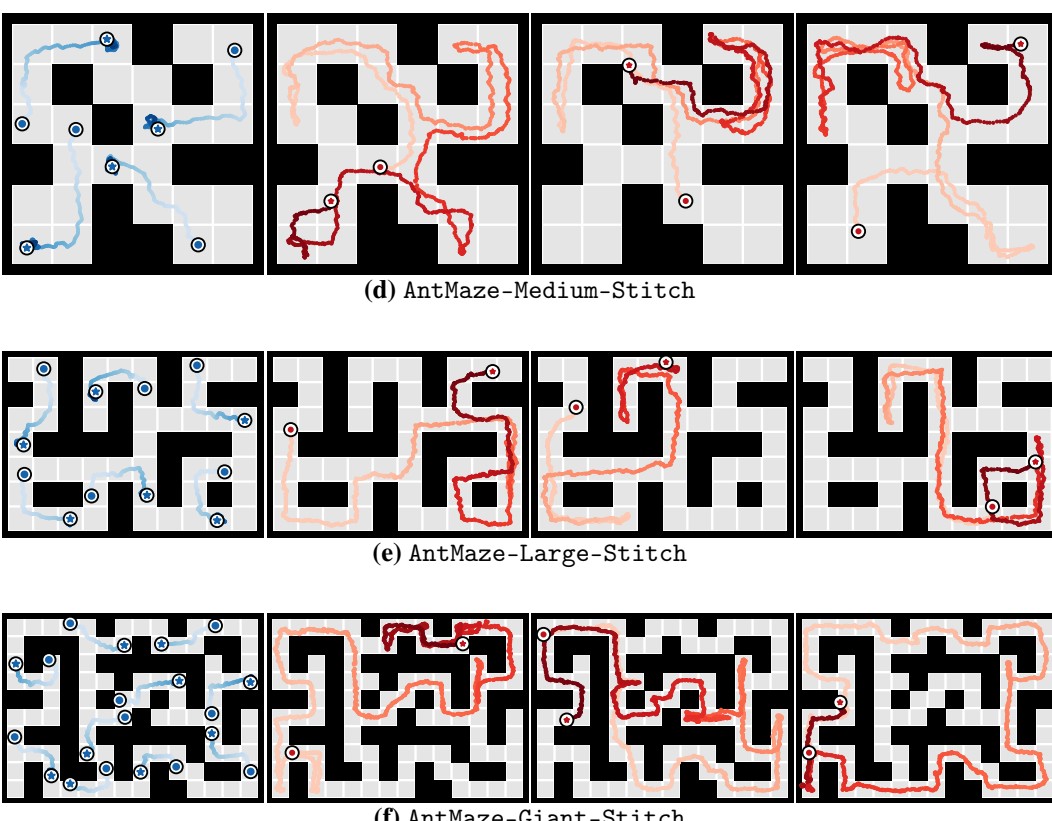

(d) `AntMaze-Medium-Stitch`

(e) `AntMaze-Large-Stitch`

(f) `AntMaze-Giant-Stitch`

Figure 12: **SCoTS-augmented trajectories for AntMaze Explore datasets.** For each AntMaze Explore dataset, the leftmost column shows trajectories from the original OGBench dataset. The subsequent columns are examples of SCoTS-generated trajectories.

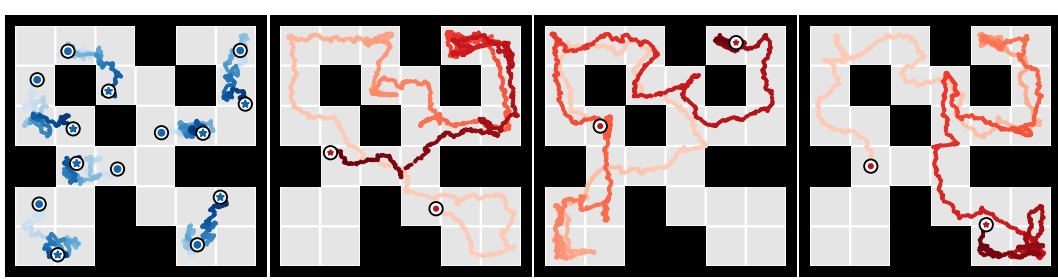

(g) `AntMaze-Medium-Explore`

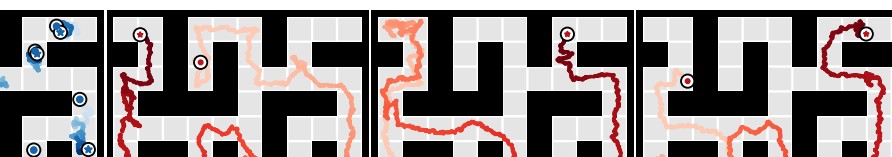

(h) `AntMaze-Large-Explore`

