# OpenReview forum: "State-Covering Trajectory Stitching for Diffusion Planners"
_NeurIPS.cc/2025/Conference — NeurIPS 2025 poster_

### Official Review · Reviewer_tqRC · 2025-06-16

**Clarity:** 4
**Significance:** 2
**Originality:** 2
**Rating:** 4
**Confidence:** 4

**Summary:**

The authors propose State-Covering Trajectory Stitching (SCoTS), a reward-free augmentation framework that generates long and diverse trajectories by incrementally stitching short segments. The method works in three stages: (1) learning a temporal-distance-preserving latent embedding to measure trajectory connectivity, (2) iteratively selecting segments based on directional progress and novelty in latent space, and (3) refining transitions between segments using a diffusion model to ensure dynamic consistency. The contribution lies in the combination of directional latent space exploration with a diffusion-based refinement pipeline for reward-free stitching.

The method is evaluated on the OGBench benchmark using PointMaze and AntMaze environments. Metrics include task success rate across various maze sizes and difficulty levels. Comparisons are made with multiple baselines, including GCIQL, HIQL, CRL, CD, and GSC, under both standard and augmented training conditions.

**Questions:**

1. How robust is SCoTS to noisy or low-quality datasets beyond the controlled OGBench settings, e.g., occupancy maps?
2. Can SCoTS be extended or adapted for online learning or continual RL settings, where the state space evolves, e.g., dynamics objects?

**Ethical Concerns:**

["NO or VERY MINOR ethics concerns only"]

**Final Justification:**

The authors have addressed all my concerns about the method's applicability in further settings. Hence, I am leaning towards accepting this paper.

**Limitations:**

Yes

**Quality:**

3

**Strengths And Weaknesses:**

**Strengths**
- **Data Augmentation Method:** SCoTS introduces a trajectory-level augmentation method tailored to diffusion planners. It cleverly combines temporal-distance-preserving embeddings, novelty-driven segment selection, and diffusion-based refinement.
- **Utility:** SCoTS-augmented data not only improves diffusion planners but also significantly boosts the performance of traditional offline goal-conditioned RL methods.
- **Reproducibility:** The paper is transparent about computational costs and issues with temporal embeddings in asymmetric or disconnected environments.

**Weaknesses**
- **Computational Overhead:** The three-stage pipeline, especially the diffusion-based refinement step, is computationally intensive. This may limit applicability in real-time or resource-constrained settings.
- **Assumption of Symmetric Temporal Embeddings:** The embedding method used may struggle in environments with highly asymmetric dynamics or irreversible transitions.
- **Limited Real-World Evaluation:** All experiments are on synthetic benchmarks. It remains unclear how well the method scales or adapts to high-dimensional real-world tasks, especially those with complex sensory inputs.
- **Evaluation Bias Toward Diffusion Planners:** While traditional GCRL methods are evaluated, the core of the paper’s benchmarks and claims primarily cater to improving diffusion planners.
- **Blending of Task-Objectives**: Given the nature of navigation robotics tasks, SCoTS could benefit from blending diffusion and gradients from task objectives as in [1], improving collision-free performance.

[1] Carvalho, Joao, et al. "Motion planning diffusion: Learning and planning of robot motions with diffusion models." 2023 IEEE/RSJ International Conference on Intelligent Robots and Systems (IROS). IEEE, 2023.

---

> ### Author Rebuttal · Authors · 2025-07-29
>
> Thank you for the detailed review and constructive feedback on our work. We especially appreciate your insightful question regarding the generalizability of SCoTS beyond the controlled OGBench setting. Please find our detailed responses below.
>
> - **“How robust is SCoTS to noisy or low-quality datasets beyond the controlled OGBench settings, e.g., occupancy maps?”**
>
> To evaluate the robustness of SCoTS on noisy and low-quality data beyond the controlled OGBench settings, we conducted additional experiments using a mobile robot navigation task in realistic, large-scale environments (a 35m x 39m corridor and a 50m x 110m building), following the setup from HRL [1].
>
> We simulated low-quality data by generating short, disconnected segments. Specifically, trajectories were collected by randomly selecting start and goal locations within short ranges (5m) using HRL [1]. To handle the noisy raw 2D LiDAR measurements (512 rays over 360°), we converted these scans into 64x64 occupancy grid images. These images were then encoded using a pre-trained $\beta$-VAE into a compact 8-dimensional latent representation, providing structured, low-dimensional state inputs for the agent. Subsequently, the SCoTS stitching method was applied to augment this fragmented dataset.
>
> The table below shows the goal-reaching success rate (over 100 episodes) of HIQL when trained with and without our SCoTS data augmentation.
>
> |  | **HIQL** | **HIQL + SCoTS (ours)** |
> | --- | --- | --- |
> | Building | 27 ± 7 | 53 ± 4 |
> | Corridor | 0 ± 0 | 31 ± 5 |
>
> These results clearly indicate that SCoTS effectively enhances performance, even with highly fragmented and noisy datasets, demonstrating its robustness and practical applicability to robot navigation tasks.
>
> - **“Can SCoTS be extended or adapted for online learning or continual RL settings?”**
>
> Although our work primarily addresses the offline setting, the SCoTS framework can naturally be extended to online or continual RL settings. One straightforward approach would involve periodically incorporating new trajectories collected by the agent into the training dataset. This evolving dataset could then be used to periodically retrain both the temporal distance-preserving embedding $\phi$ and the diffusion-based stitcher $p_{\theta}^{\text{stitcher}}$. This strategy aligns with successful online data augmentation approaches demonstrated in prior work [2]. We believe further extending the idea behind SCoTS to other setting is an exciting future research direction.
>
> - “**Computational overhead”**
>
> We acknowledge that SCoTS introduces additional computational overhead, primarily due to the training of the diffusion-based stitcher and the trajectory augmentation process, which together roughly equate to 50% of the training time required for the final diffusion planner.
>
> However, we believe this overhead is justified by substantial performance gains. For instance, on challenging tasks such as antmaze-explore-giant, SCoTS improved the hierarchical diffusion planner's success rate by up to seven times. Such considerable improvement in generalization, achieved through a single offline augmentation phase, is highly beneficial, particularly when compared to the significantly higher costs associated with collecting a diverse set of high-quality, long-horizon data.
>
> - “**Assumption of Symmetric Temporal Embeddings”**
>
> As we discussed in our paper's limitations section, our temporal distance-preserving embedding assumes symmetric temporal embeddings. We believe this limitation can be resolved by leveraging universal quasimetric embedding as explored in [3] which allows the representation to more faithfully model irreversible dynamics, which we leave for future work.
>
> - **“Evaluation bias toward diffusion planners”**
>
> While our motivation and main narrative center around enhancing diffusion planners, we emphasize that our experimental results show substantial improvements for traditional offline goal-conditioned RL (GCRL) algorithms as well.
>
> Specifically, our work is the first, to our knowledge, to enable GCRL methods to surpass 50% success rates on particularly challenging OGBench tasks (antmaze-stitch-giant and antmaze-explore-large). These results demonstrate that SCoTS serves as a general-purpose data augmentation framework that significantly improves performance across diverse offline RL algorithms, not merely diffusion-based approaches.
>
> We believe these results strengthen rather than weaken our contribution.
>
> - “**Given the nature of navigation robotics tasks, SCoTS could benefit from blending diffusion and gradients from task objectives as in [4], improving collision-free performance.**”
>
> We thank the reviewer’s suggestion. We would like to clarify that our SCoTS framework and the suggested guidance mechanism are orthogonal and complementary. SCoTS is a data augmentation method that enriches the training data to provide the planner with a stronger, more diverse prior. In contrast, the guidance technique is an inference-time method that refines a generated plan to satisfy specific constraints like collision avoidance. We believe this combination is an interesting direction for future research and will add a discussion of this potential synergy in the final version of the paper.
>
> References:
>
> [1] Lee et al., Adaptive and Explainable Deployment of Navigation Skills via Hierarchical Deep Reinforcement Learning. ICRA, 2023.
>
> [2] Lu et al., Synthetic Experience Replay. NeurIPS, 2023.
>
> [3] Wang et al., Optimal goal-reaching reinforcement learning via quasimetric learning. ICML, 2023.
>
> [4]  Carvalho et al., Motion planning diffusion: Learning and planning of robot motions with diffusion models. IROS, 2023.

---

> > ### Comment · Reviewer_tqRC · 2025-08-02
> >
> > The author nicely addressed all my concerns about the method's applicability in different settings. Hence, I raised my score.

---

> > > ### Author Response · Authors · 2025-08-02
> > >
> > > We sincerely appreciate your time and dedication to providing us with your valuable feedback.

---

### Official Review · Reviewer_nESC · 2025-06-17

**Clarity:** 3
**Significance:** 2
**Originality:** 2
**Rating:** 5
**Confidence:** 4

**Summary:**

This paper proposed a novel, reward-free method for trajectory stitching in offline long-horizon planning. Unlike previous method, this method performs planning in a temporal distance-preserving embedding space, and encourages the novelty of sub-trajectory generation to make the exploration of whole state spaces. Empirical results are provided to support the claim for the proposed method.

**Questions:**

see weakness

**Ethical Concerns:**

["NO or VERY MINOR ethics concerns only"]

**Final Justification:**

All my concerns in the review have been well addressed by the authors, so I suggest the acceptance for this paper.

**Limitations:**

Yes

**Paper Formatting Concerns:**

No major formatting issues.

**Quality:**

3

**Strengths And Weaknesses:**

**strength:** The paper is well-structed and easy to follow. The idea is well-motivated and the empirical analysis is sufficient to demonstrate the effectiveness of proposed method.

**weakness:**

1. The learning of Temporal Distance-Preserving Embedding. One major concern for this part is the convergence and optimality of the learning of this embedding. That is, by the loss function provided by Eq.(7), what kind of embedding function $\phi$ will be outputted? Can it actually learns the claimed embedding of optimal temporal distance $d^*(s,g)$? More evidences (better from theoretical aspect) should be provided.

2. (Bias and Generalization ability) Following weakness 1, if the learned embedding is biased, how will it affect the subsequent process and final performance? Even if we can successfully find the neighbor states by Eq.(8), how can it guarantee that the inverse dynamics model is able to successfully predict the action to combine the two states (end($\tau_{comp}$), initial($\tau_{best}$)), if this state pair is never demonstrated in the dataset.

3. Some details are not clear. For instance, the formulation of TopKNeighbours() function in Eq.(8) is not given; Section 3.3 is not well-structured, where the stitcher should be more detailed formulated and how can it avoid to 'exhibit minor dynamic inconsistencies or sub-optimal transitions'? Furthermore, all the offline RL agents should be guided by reward, so how does reward affect the agent in this method?

4. Ablation study and sensitive analysis are not sufficient. There should be many hyperparameters in this method that will significantly affect the performance, i.e., the length of sub-trajectories, the K in Eq.(8) and (10), and the $\beta$ in Eq.(11). However, their sensitivities are not well analyzed.

5. (Optional) We are happy to see the results on other offline RL benchmarks such as D4RL (e.g., MuJoCo), to demonstrate the effectiveness of the proposed method on a broader scope.

**Additional comments:** In my opinion, this work is potential. Although a borderline accept is given currently, I am open for further discussion and eager to see the improvements during rebuttal.

---

> ### Author Rebuttal · Authors · 2025-07-29
>
> Thank you for the detailed review and constructive feedback on our work. We especially appreciate your helpful comments and suggestions for further improving the work. Please find our detailed responses below.
>
> - **“Can $\phi$ actually learns the embedding of optimal temporal distance?”**
>
> Learning temporal embedding $\phi$ is grounded in the well-established equivalence between the optimal temporal distance and the optimal goal-conditioned value function [1, 2, 3]: $V^{\*}(\boldsymbol{s},\boldsymbol{g}) = -d^{\*}(\boldsymbol{s},\boldsymbol{g})$
>
> Here, $V^{\*}(\boldsymbol{s},\boldsymbol{g})$ is the maximum possible cumulative sum of rewards for setting where an agent receives a reward of $0$ if the $l_2$ distance between states $\boldsymbol{s}$ and $\boldsymbol{g}$  is within a small threshold $\delta_g$, and $-1$ otherwise. We leverage this by parameterizing our value function directly as a distance in the latent space: $V(\boldsymbol{s}, \boldsymbol{g}) \coloneqq -||\phi(\boldsymbol{s}) - \phi(\boldsymbol{g})||_2$
>
> By training $\phi$ to minimize a standard IQL-based temporal difference (TD) error for this value function, we make $V(\boldsymbol{s}, \boldsymbol{g})$  to approximate the true optimal value $V^{\*}(\boldsymbol{s}, \boldsymbol{g})$ . Consequently, this implicitly forces our learned latent distance, $||\phi(\boldsymbol{s}) - \phi(\boldsymbol{g})||_2$ to approximate the optimal temporal distance, $d^{\*}(\boldsymbol{s},\boldsymbol{g})$. Empirically, this approximation proves effective, though we acknowledge it remains an approximation.
>
> - **“How will embedding error affect the subsequent process and final performance?”**
>
> Thank you for this important question. The robustness of our method to imperfections in the learned embedding is a core design consideration. We explicitly acknowledge that the embedding $\phi$ can be a biased approximation, and SCoTS incorporates two key mechanisms to mitigate the potential negative effects during the stitching process.
>
> First, embedding errors can lead to the retrieval of candidate segments that are not dynamically consistent with the current trajectory. This can result in stitches with large dynamic inconsistency errors at the connection points. To address this, we employ our Diffusion-based Stitching Refinement step, which is specifically designed to smooth the transition between segments and ensure dynamic feasibility, as shown in Figure 7 of our Appendix.
>
> To demonstrate its necessity, we ablate this component. The results show that removing refinement significantly degrades performance for both the diffusion planner (HD) and, more dramatically, the GCRL algorithm (HIQL), which is highly sensitive to the dynamic validity of transitions.
>
> |  | **HD + SCoTS (w/o refinement)** | **HD + SCoTS (w/ refinement)** | **HIQL + SCoTS (w/o refinement)** | **HIQL + SCoTS (w/ refinement)** |
> | --- | --- | --- | --- | --- |
> | antmaze-stitch-large | 85 ± 3 | 93 ± 1 | 52 ± 2 | 91 ± 2 |
> | antmaze-stitch-giant | 53 ± 1 | 87 ± 2 | 11 ± 2 | 55 ± 5 |
>
> Second, geometric distortions in the latent space can make a straight-line exploration path suboptimal. Consequently, a straight line between two latent states, $\phi(s)$ and $\phi(s')$, may not correspond to a feasible path in the underlying MDP. Therefore, exploratory detours are often necessary to find a valid temporally extended path. This is precisely where the novelty score becomes critical. It encourages these exploratory detours by rewarding the selection of segments that lead to less-visited states. This allows the agent to navigate around the imperfections in the latent space, discovering more temporally extended paths.
>
> The following ablation study on the novelty weight $\beta$ demonstrates this complementary effect. The results show that while directional stitching alone $(\beta=0)$ significantly outperforms the baseline, introducing a novelty score $(\beta>0)$ provides substantial further gains.
>
> |  | **HD** | **beta=0** | **beta=1** | **beta=2** | **beta=4** | **beta=8** |
> | --- | --- | --- | --- | --- | --- | --- |
> | antmaze-stitch-large | 36 ± 2 | 87 ± 2 | 94 ± 1 | 93 ± 1 | 92 ± 1 | 85 ± 3 |
> | antmaze-stitch-giant | 0 ± 0 | 63 ± 3 | 72 ± 2 | 87 ± 2 | 89± 2 | 74 ± 3 |
> - **“Formulation of TopKNeighbours”**
>
> The function $\text{TopKNeighbors}(\text{query}, \text{search space}, k)$ is a standard k-nearest neighbor search algorithm. In the context of our paper, its query is the latent embedding of the final state of the currently composed trajectory, $\phi(\text{end}(\boldsymbol{\tau}_{\text{comp}}))$, and the search space is the the set of latent embeddings of the initial states of all trajectories within the entire offline dataset, denoted by the shorthand $\phi(\mathcal{D})$.
>
> The function operates by searching this space to find the $k$ vectors closest to the query, using the Euclidean distance in the learned latent space as the metric. It then returns the $k$ original trajectory segments from the dataset that correspond to these nearest initial states, which form the candidate set for the next stitching step. (Practical implementation can be found in our Appendix B.)
>
> We appreciate the reviewer highlighting this ambiguity and will clarify this point carefully in the final version of the paper.
>
> - **“How can diffusion-based stitching refinement avoid to “exhibit minor dynamic inconsistencies or sub-optimal transitions”?**
>
> The key to avoiding inconsistencies lies in the fact that our method is generative, not simple concatenation. A naive approach would be to simply concatenate two segments, which could create an abrupt, or even dynamically impossible transition at the stitching point. Our diffusion-based stitcher avoids this by generating an entirely new bridging path. Because the stitcher is a diffusion model trained on the entire dataset of real trajectories, it learns a strong prior over the distribution of physically plausible state sequences.  When conditioned on the boundary states ($\boldsymbol{s}_1$ and  $\boldsymbol{s}_H$), it samples a trajectory from the learned distribution of valid trajectories.
>
> - **“Ablation study and sensitivity analysis.”**
>
> Thank you for your valuable feedback. Following the suggestion, we have conducted additional ablation and sensitivity analyses to better understand the impact of key hyperparameters and the necessity of our core components.
>
> **1. Sensitivity to Sub-trajectory Length $H$**
>
> |  | **H=26** | **H=52** | **H=104** |
> | --- | --- | --- | --- |
> | antmaze-stitch-giant | 87 ± 2 | 89 ± 2 | 87 ± 2 |
> | antmaze-explore-large | 98 ± 1 | 93 ± 1 | 88 ± 2 |
>
> The results show that on the stitch dataset, performance is relatively robust to the choice of $H$. However, on the explore dataset, shorter segments perform best. We hypothesize this is because the explore dataset consists of low-quality, noisy trajectories. Using shorter segments allows SCoTS to be more selective, finding and connecting the temporally-extended parts of trajectories.
>
> **2. Sensitivity to Number of Retrieved Segments $K$**
>
> |  | **K=3** | **K=10** | **K=20** |
> | --- | --- | --- | --- |
> | antmaze-stitch-giant | 65 ± 3 | 87 ± 2 | 89 ± 2 |
> | antmaze-explore-large | 72 ± 2 | 98 ± 1 | 97 ± 1 |
>
> Performance is robust as long as $K$ is not too small. A very small $K$ limits the diversity of candidate segments, hindering the effectiveness of our progress and novelty-based selection. While performance is stable for larger $K$, we expect an excessively large $K$ could eventually degrade performance by increasing the chance of retrieving dynamically inconsistent segments.
>
> **3. Sensitivity to Novelty Weight $\beta$**
>
> We respectfully refer the reviewer to our detailed response to second question.
>
> **4. Necessity of Core Components**
>
> They are discussed in the rebuttal for reviewer RD6p due to the space limitation.
>
> **“We are happy to see the results on other offline RL benchmarks”** and **“How does reward affect the agent in this method?”**
>
> While our paper focuses on tasks where extrinsic rewards are absent, the SCoTS framework is not limited to goal-conditioned tasks.
>
> To clarify, reward-free refers to our stitching mechanism, which is guided by the intrinsic temporal structure rather than extrinsic task rewards. This allows the core data augmentation to be task-agnostic. To demonstrate its applicability to general, dense-reward offline RL, we conducted additional experiments on four standard MuJoCo locomotion benchmarks from D4RL, utilizing suboptimal medium and medium-replay datasets.
>
> For a fair comparison, we used the same Diffuser architecture and planning horizon (32) as described in the original paper. We upsample the original data to 2M samples using the same SCoTS procedure, with sub-trajectory lengths set to 4. Additionally, to label these new trajectories with rewards, we additionally trained a reward model, $r_t = f_{\omega}(s_t,a_t)$, on the original offline dataset. The table below shows the normalized average returns over 50 planning seeds.
>
> |  | **Diffuser** | **Diffuser + SCoTS (ours)** |
> | --- | --- | --- |
> | hopper-medium | 74.3 ± 1.4 | 93.5 ± 1.21 |
> | walker2d-medium | 79.6 ± 0.55 | 80.2 ± 0.5 |
> | hopper-medium-replay | 93.6 ± 0.4 | 96.7 ± 0.62 |
> | walker2d-medium-replay | 70.6 ± 1.6 | 78.5 ± 1.86 |
> | **average** | 79.5 | **87.2** |
>
> The results show that training on SCoTS-augmented data consistently improves the performance of the original Diffuser. Even without architectural or sampling strategy changes, by stitching sub-trajectories in a way that is temporally extended and exploratory, SCoTS generates novel behaviors (e.g., trajectories that travel further) that are not present in the original suboptimal dataset.
>
> References:
>
> [1] Kaelbling et al., . Learning to achieve goals. IJCAI, 1993.
>
> [2] Wang et al., Optimal goal-reaching reinforcement learning via quasimetric learning. ICML, 2023.
>
> [3] Park et al., Foundation Policies with Hilbert Representations. ICML, 2024.

---

> > ### Comment · Reviewer_nESC · 2025-08-03
> > **Response to rebuttal (R1)**
> >
> > Thanks for the authors' detailed rebuttals, which have addressed most of my concerns.
> >
> > However, about weakness 1, I think more evidences should be provided, for the following reasons,
> >
> > 1. The core contribution of this work is based on the temproal distance-preserving latent space, so it is critical to carefully demonstrate and discuss about its effectiveness.
> >
> > 2. The learning of the temproal distance-preserving latent space has its natural limitations [1], as:
> >     2.1) The learned distance metric in latent space is symmetric, while the groundtruth might be asymmetric.
> >     2.2) There may not exists exist an exact isometry between MDP and the Hilbert space.
> >     2.3) A discount factor \gamma is used in learning, but it does not exist in practice.
> > As a result, the learned latent space would be naturally biased to the claims.
> >
> > [1] Park S, Kreiman T, Levine S. Foundation Policies with Hilbert Representations[C]. International Conference on Machine Learning. PMLR, 2024: 39737-39761.
> >
> > To summary, what I really care about is that **what kind of latent space you have learned finally, and why do you think it is reliable enough for the offline trajectory stitching tasks?**. Thanks!

---

> > > ### Author Response · Authors · 2025-08-03
> > >
> > > We sincerely thank reviewer nESC for going through our response.
> > >
> > > To clarify, the latent space we have ultimately learned is not a perfect metric representation of the MDP. Instead, it is a latent space in which temporal proximity in the original MDP is correlated with Euclidean proximity in this learned latent space. As you correctly pointed out, this learned space has inherent limitations (a symmetric metric vs. asymmetric dynamics, the lack of a guaranteed isometry, and discount factor bias).
> > >
> > > Our central argument for its reliability, despite these imperfections, is that **SCoTS sidesteps the need for a globally accurate temporal distance.** Instead, it leverages the latent distance for a much less demanding and more practically achievable task: solving the **local problem** at each step of the iterative stitching process. Specifically, it retrieves a small set of promising candidate segments that are locally reachable and identifies segments that extend the trajectory locally in time. Solving this local problem is significantly more tractable and robust to embedding imperfections than relying on perfectly accurate global temporal distances.
> > >
> > > Thank you again for your valuable feedback. Please let us know if there is anything else we can clarify.

---

> > > > ### Comment · Reviewer_nESC · 2025-08-04
> > > > **Final response**
> > > >
> > > > Thanks for the authors response. I suggest the authors add all the contents during the rebuttal into the revised paper. Then since all the concerns have been addressed, I have raised my score to 5. Good luck.

---

> > > > > ### Author Response · Authors · 2025-08-04
> > > > >
> > > > > We sincerely appreciate your insightful and thoughtful comments, and the time you took to review our paper.

---

### Official Review · Reviewer_RD6p · 2025-07-01

**Clarity:** 3
**Significance:** 3
**Originality:** 3
**Rating:** 5
**Confidence:** 4

**Summary:**

This paper introduces State-Covering Trajectory Stitching (SCOTS), a method to improve diffusion-based planners that are fundamentally limited by the diversity and horizon of their offline training data. SCOTS addresses this by augmenting datasets with diverse, long-horizon trajectories. The framework first learns a temporal distance-preserving latent representation of the environment's states. It then iteratively stitches short trajectory segments together, selecting segments that balance progress along a random exploratory direction with novelty to effectively expand state-space coverage. A diffusion-based model refines the connection points between segments to ensure the final trajectories are dynamically consistent. Experiments demonstrate that training on SCOTS-augmented data significantly enhances the generalization and long-horizon planning capabilities of diffusion planners. The method also provides a notable performance boost to widely-used offline goal-conditioned reinforcement learning algorithms.

**Questions:**

1. Figure 1 in the paper demonstrates that the current Diffuser method fails to generalize well to these out-of-distribution tasks. However, Figure 3.b in the original Diffuser paper (https://arxiv.org/pdf/2205.09991) seems to show that it can achieve a similar trajectory stitching effect to the proposed method. How do the authors explain this apparent contradiction?

2. Why is it necessary to balance the randomly sampled latent direction with the novelty relative to previously visited states?

**Ethical Concerns:**

["NO or VERY MINOR ethics concerns only"]

**Final Justification:**

The authors provided a thoughtful and comprehensive rebuttal that effectively addressed all my initial concerns. Their clarifications resolved key questions about the method's design and motivation, and the new experiments — particularly on offline RL benchmarks and through ablation studies — provide strong empirical support.

Concerns raised by other reviewers, such as computational cost, were also addressed with reasonable explanations. While the original experimental scope was limited, the additional results broaden the applicability and reinforce the contributions.

Based on these updates, I have increased my score and recommend acceptance. I encourage the authors to include the new results in the final version to further strengthen the paper.

**Limitations:**

See the first point in the Weaknesses section.

**Paper Formatting Concerns:**

No Paper Formatting Concerns.

**Quality:**

3

**Strengths And Weaknesses:**

Strengths

1. This work demonstrates significant performance improvements over the compared baselines on the OGBench benchmark.

Weaknesses

1. The paper emphasizes that the proposed method is *reward-free* as a key distinction from prior works. However, there is no analysis or discussion throughout the paper to explain the importance or effectiveness of this reward-free property. In fact, if I understand correctly, if the augmented data does not contain rewards, the method would not be effective for most offline RL tasks and would only be applicable to specific goal-conditioned tasks, which further limits the applicability of this approach.

2. The writing is sometimes confusing, and several claims or proposed solutions lack justification—for example, through illustrative toy experiments or empirical evidence on real tasks. For instance: why is it necessary to balance the randomly sampled latent direction with novelty relative to previously visited states? Aren’t both intended to enhance exploration?

3. The proposed method incorporates many components from prior works and several technical mechanisms, which leans toward an engineering-heavy design—which is fine. However, the paper does not sufficiently demonstrate, even through ablation studies, the necessity of including these components. I suggest adding experiments to support this aspect.

---

> ### Author Rebuttal · Authors · 2025-07-29
>
> Thank you for the detailed review and constructive feedback on our work. We especially appreciate your insightful question about the necessity of balancing directional progress with exploratory novelty in our trajectory stitching method, which is particularly helpful for clarifying our contribution. Please find our detailed responses below.
>
> - **“Contradiction between our Fig.1 and Diffuser original paper's Fig.3.b?”**
>
> Thanks for asking this valid question. The success shown in the original Diffuser paper's Figure 3.b was on a relatively simple toy experiments. In such settings, standard U-Net-based diffusion models can perform well. However, this architectural property becomes a critical limitation in the long-horizon and high-dimensional OOD settings we target. We hypothesize this is because the aggressive downsampling from pooling and strided convolutions reduces the bottleneck representation so drastically that even small kernels can access distant sequence information, a property that is helpful in simple settings but detrimental in complex ones where temporal details are lost.
>
> This limitation has been recognized issue in the community, as acknowledged in prior works [2, 3]. For example, the appendix of [3] presents a similar analysis and empirically illustrates this failure mode (Figure 4 in Appendix). Therefore, we argue there is no contradiction. The original work demonstrated success in a constrained setting, while our work and Figure 1 address the failure of these models to generalize to more challenging scenarios.
>
> - **“Why balance directional progress with novelty?”**
>
> As illustrated in our paper's Figure 4, using only the progress score ($\beta=0$), which quantifies the alignment with a latent exploration direction, is effective at generating long, temporally-extended trajectories with clear directional distinctions.
>
> However, relying solely on this directional guidance can be suboptimal. The learned temporal distance-preserving embedding $\phi$ is a powerful but imperfect approximation of the true environment topology. Due to embedding errors, the geometry of the latent space can be distorted. Consequently, a straight line between two latent states, $\phi(s)$ and $\phi(s')$, may not correspond to a feasible path in the underlying MDP. Therefore, exploratory detours are often necessary to find a valid temporally extended path.
>
> This is precisely where the novelty score becomes critical. It encourages these exploratory detours by rewarding the selection of segments that lead to less-visited states. This allows the agent to navigate around the imperfections in the latent space, discovering more temporally extended paths.
>
> To demonstrate this complementary effect quantitatively, we conducted an additional ablation study on 2 OGBench tasks. The table below compares the performance of the HD planner trained with SCoTS data generated using different novelty weights $\beta$ (5 seeds).
>
> |  | **HD** | **beta=0** | **beta=1** | **beta=2** | **beta=4** | **beta=8** |
> | --- | --- | --- | --- | --- | --- | --- |
> | antmaze-stitch-large | 36 ± 2 | 87 ± 2 | 94 ± 1 | 93 ± 1 | 92 ± 1 | 85 ± 3 |
> | antmaze-stitch-giant | 0 ± 0 | 63 ± 3 | 72 ± 2 | 87 ± 2 | 89± 2 | 74 ± 3 |
>
> The results suggest that while directional stitching alone ($\beta=0$) provides a notable performance improvement, introducing and balancing it with the novelty score ($\beta>0$) yields substantial further gains. This effect is especially pronounced on the more complex Giant task.
>
> - **Applicability of SCoTS to general offline RL.**
>
> While our paper focuses on tasks where extrinsic rewards are absent, the SCoTS framework is not limited to goal-conditioned tasks.
>
> To clarify, reward-free refers to our stitching mechanism, which is guided by the intrinsic temporal structure rather than extrinsic task rewards. This allows the core data augmentation to be task-agnostic. To demonstrate its applicability to general, dense-reward offline RL, we conducted additional experiments on four standard MuJoCo locomotion benchmarks from D4RL, utilizing suboptimal medium and medium-replay datasets.
>
> For a fair comparison, we used the same Diffuser architecture and planning horizon (32) as described in the original paper. We upsample the original data to 2M samples using the same SCoTS procedure, with sub-trajectory lengths set to 4. Additionally, to label these new trajectories with rewards, we additionally trained a reward model, $r_t = f_{\omega}(s_t,a_t)$, on the original offline dataset. The table below shows the normalized average returns over 50 planning seeds.
>
> |  | **Diffuser** | **Diffuser + SCoTS (ours)** |
> | --- | --- | --- |
> | hopper-medium | 74.3 ± 1.4 | 93.5 ± 1.21 |
> | walker2d-medium | 79.6 ± 0.55 | 80.2 ± 0.5 |
> | hopper-medium-replay | 93.6 ± 0.4 | 96.7 ± 0.62 |
> | walker2d-medium-replay | 70.6 ± 1.6 | 78.5 ± 1.86 |
> | **average** | 79.5 | **87.2** |
>
> The results show that training on SCoTS-augmented data significantly and consistently improves the performance of the original Diffuser. Even without architectural or sampling strategy changes, by stitching sub-trajectories in a way that is temporally extended and exploratory, SCoTS generates novel behaviors (e.g., trajectories that travel further) that are not present in the original suboptimal dataset.
>
> - **“Additional experiments to demonstrate the necessity of each component.”**
>
> We appreciate the reviewer's valuable suggestion. We have conducted additional ablation studies to empirically demonstrate the necessity of each key component of SCoTS.
>
> To isolate the contribution of our learned latent space, we compare the performance of the HD planner trained on SCoTS-augmented data where the stitching process was guided by either our temporal distance-preserving embedding or raw state space.
>
> |  | **HD + SCoTS (w/o temporal embedding)** | **HD + SCoTS (w/ temporal embedding)** |
> | --- | --- | --- |
> | pointmaze-stitch-large | 93 ± 0 | 100 ± 0 |
> | pointmaze-stitch-giant | 52 ± 1 | 100 ± 0 |
> | antmaze-stitch-large | 45 ± 1 | 93 ± 1 |
> | antmaze-stitch-giant | 7 ± 2 | 87 ± 2 |
>
> The results clearly show that the temporal distance-preserving representation is crucial for effective stitching. This is especially true in high-dimensional environments like antmaze, where a small Euclidean distance in the raw state space (e.g., between two similar joint configurations) does not guarantee reachability. Even in pointmaze, where state-space distance is more intuitive, the learned compact latent space provides a better-structured representation for stitching temporally extended trajectories.
>
> To demonstrate the importance of the refinement step, we compare the performance of both a diffusion planner (HD) and a GCRL algorithm (HIQL) trained on SCoTS data generated with and without the diffusion-based refinement.
>
> |  | **HD + SCoTS (w/o refinement)** | **HD + SCoTS (w/ refinement)** | **HIQL + SCoTS (w/o refinement)** | **HIQL + SCoTS (w/ refinement)** |
> | --- | --- | --- | --- | --- |
> | antmaze-stitch-large | 85 ± 3 | 93 ± 1 | 52 ± 2 | 91 ± 2 |
> | antmaze-stitch-giant | 53 ± 1 | 87 ± 2 | 11 ± 2 | 55 ± 5 |
>
> These results reveal the critical importance of the refinement component. Without refinement, the connection points between stitched segments can suffer from large dynamic inconsistency errors, as illustrated in Figure 7 of our appendix. This performance degradation is particularly significant when training GCRL algorithms like HIQL, which are highly sensitive to the dynamic validity of the training transitions.
>
> References:
>
> [1] Janner et al., Planning with Diffusion for Flexible Behavior Synthesis. ICML, 2022.
>
> [2] Chen et al., Diffusion Forcing: Next-token Prediction Meets Full-Sequence Diffusion. NeurIPS, 2024.
>
> [3] Chen et al., Simple Hierarchical Planning with Diffusion. ICLR, 2024.

---

> > ### Author Response · Authors · 2025-08-05
> >
> > Dear Reviewer RD6p,
> >
> > We sincerely apologize for any inconvenience caused by this reminder. We just wanted to kindly remind you that the discussion period ends in 2 days.
> >
> > As the discussion period is nearing its end, we would greatly appreciate it if you could let us know if you have any remaining questions or suggestions. We again appreciate your time and dedication to providing us with your valuable feedback.
> >
> > Best,
> >
> > The Authors

---

> > ### Comment · Reviewer_RD6p · 2025-08-07
> >
> > Thank you for the thoughtful and comprehensive rebuttal. I appreciate the authors’ detailed clarifications on Questions 1 and 2 as well as Weakness 2 — they have resolved my concerns, and I have no further questions.
> >
> > Regarding Weaknesses 1 and 3, the authors have provided convincing experimental evidence to demonstrate the effectiveness of the proposed method, including both its application to offline RL algorithms and ablation studies. I find the results compelling and believe they successfully address these weaknesses.
> >
> > While the original paper’s experimental settings were somewhat narrow and limited the demonstration of generality across environments, the additional experiments on D4RL in the rebuttal partially mitigate this concern.
> >
> > Based on these clarifications and new results, I now believe that the contributions are sufficiently solid and impactful to warrant acceptance. I have increased my score to 5 accordingly and would encourage the authors to **incorporate these additional experiments into the final version** to further strengthen the paper.

---

> > > ### Author Response · Authors · 2025-08-07
> > >
> > > We sincerely appreciate your time and thoughtful feedback, which helps us further improve this work.

---

### Official Review · Reviewer_C6Zi · 2025-07-03

**Clarity:** 3
**Significance:** 3
**Originality:** 3
**Rating:** 5
**Confidence:** 3

**Summary:**

This work proposes a novel method called State-Covering Trajectory Stitching (SCoTS) for trajectory augmentation. The method first learns a temporal distance-preserving embedding that encodes states into a latent representation. Then, it randomly samples trajectory segments and selects candidate segments to concatenate based on the proposed progress score and novelty score. The resulting augmented trajectories are used to supplement the original offline dataset for training the final diffusion planner. Experimental results demonstrate that the method performs well on long-horizon tasks.

**Questions:**

1. In the remark paragraph of the Planning with Diffusion Models section, the authors claimed that the augmented dataset $\mathcal{D}_\text{aug}$ would extend trajectory coverage. However, as far as I understand, the augmented data does not generate new states but rather concatenates different segments of existing trajectories. Does this really bring substantial benefits compared to other diffusion planning methods that decompose trajectories into subgoals, such as Monte Carlo Tree Diffusion [1] or Hierarchical Diffuser [2]?

2. In Section 3.2, are the candidate trajectory segments $\tau_j$ sampled from the same original trajectories, or are they drawn across different trajectories?

[1] Yoon, Jaesik, et al. Monte Carlo Tree Diffusion for System 2 Planning. Proceedings of the Forty-second International Conference on Machine Learning, 2025.

[2] Chen, Chang, et al. Simple Hierarchical Planning with Diffusion. The Twelfth International Conference on Learning Representations, 2024.

**Ethical Concerns:**

["NO or VERY MINOR ethics concerns only"]

**Final Justification:**

I thank the authors for the detailed rebuttal and their prompt responses to the follow-up discussions. The authors have thoroughly discussed their contributions in comparison to prior works. Although I remain somewhat unclear about the contributions relative to other stitching-based planning methods, I find the paper solid and the method promising. I have increased my score by 1.

**Limitations:**

The limitations of this work are discussed in the conclusion section.

**Paper Formatting Concerns:**

There are no major formatting issues in this paper.

**Quality:**

3

**Strengths And Weaknesses:**

Strengths:

1. The idea of introducing trajectory stitching to generate augmented data using both the proposed progress score and novelty score is innovative.

2. Comprehensive experimental results on multiple tasks demonstrate the effectiveness of the work and prove that the evaluation is sufficient.

3. Concrete details of the implementation are provided in the appendix to ensure the reproducibility of this work.

Weaknesses:

1. Lacking theoretical results to prove the correctness of the methods, especially in the directional and exploratory trajectory stitching part. The idea is interesting, but there is no guarantee that stitching different trajectory segments would help planning.

2. Some minor issues:

   - In the first paragraph of Section 3.2, there seems to be no difference between the unit vector $z$ and the vector itself in $z \gets \frac{z}{∥z∥}$.

   - The state encoder $\phi(s)$ should take the state as input, but in Eq. (8), $\phi(\mathcal{D})$ appears to take the dataset $\mathcal{D}$ as input. The authors may want to use an alternative representation.

---

> ### Author Rebuttal · Authors · 2025-07-29
>
> Thank you for the detailed review and constructive feedback on our work. We especially appreciate your insightful question comparing the benefits of our trajectory stitching approach against subgoal decomposition methods, which is particularly helpful for clarifying our contribution.  Please find our detailed responses below.
>
> - **“Does SCoTS really bring substantial benefits compared to methods like MCTD [1] and HD [2]?**”
>
> Thank you for this important question. We would like to clarify the fundamental difference between our approach and prior methods such as MCTD [1] and HD [2].
>
> The fundamental limitation of these methods is that their planning horizon is inherently constrained by the maximum length of trajectories seen during training. Consequently, they typically struggle with tasks requiring extensive stitching, such as those in the OGBench *Stitch* dataset, where individual trajectories are limited to 200 steps but successful task completion requires trajectories of 800+ steps.
>
> In contrast, SCoTS systematically generates diverse, extended trajectories, effectively breaking the horizon limitation of existing planners. To demonstrate this empirically, we compare SCoTS with MCTD [1] and HD [2] on PointMaze environments with two dataset types:
>
> - Navigate: Contains trajectories long enough to directly solve test tasks.
> - Stitch: Contains short trajectories (max 4 cells), requiring extensive stitching.
>
> |  | **MCTD [1]** | **HD [2]** | **HD + SCoTS (ours)** |
> | --- | --- | --- | --- |
> | pointmaze-navigate-medium | 100 ± 0 | 100 ± 0 | 100 ± 0 |
> | pointmaze-navigate-large | 98 ± 6 | 87 ± 3 | 100 ± 0 |
> | pointmaze-stitch-medium | 90 ± 14 | 24 ± 3 | 100 ± 0 |
> | pointmaze-stitch-large | 20 ± 0 | 17 ± 2 | 100 ± 0 |
>
> As shown, methods like HD and MCTD struggle significantly when extensive trajectory stitching is required. In contrast, HD trained with SCoTS-augmented data achieves consistent success, demonstrating our method's clear benefit.
>
> - **“Are the candidate trajectory segments sampled from the same original trajectories, or are they drawn across different trajectories?”**
>
> To clarify, the candidate trajectory segments $\boldsymbol{\tau}_j$ sampled are drawn from across all different trajectories within the entire offline dataset $\mathcal{D}$, not from the same original trajectory.
>
> This is explicitly handled by our search mechanism described in Section 3.2 and Equation (8): $\text{TopKNeighbors}(\phi(\text{end}(\boldsymbol{\tau}_{\text{comp}})),\phi(\mathcal{D}),k).$
>
> Here, the search space $\phi(\mathcal{D})$ consists of the latent embeddings of the initial states of all trajectories in the offline dataset. Our method finds the nearest neighbors to the current endpoint $\text{end}(\boldsymbol{\tau}_{\text{comp}})$ from this global pool of candidates.
>
> This cross-trajectory stitching is fundamental to effectiveness of SCoTS. By connecting segments from entirely different source trajectories, we can synthesize novel behavioral sequences that were never demonstrated together in the original data. This enables the generation of diverse, extended trajectories that explore previously unseen regions of the trajectory space, significantly enhancing the diffusion planner's generalization capabilities beyond the original data distribution.
>
> We appreciate the reviewer highlighting this ambiguity and will clarify this point carefully in the final version of the paper.
>
> - **“How directional and exploratory trajectory stitching would help planning.”**
>
> The rationale behind our stitching strategy is inspired by recent successes in unsupervised reinforcement learning [3, 4, 5], demonstrating that state-covering behaviors significantly benefit downstream tasks.
>
> Our work addresses a critical limitation in offline RL: datasets are often composed of disconnected, short-horizon, or narrow-distribution trajectories. Consequently, the strategy for how to stitch these trajectories becomes a crucial factor. For example, a naive stitching approach, such as randomly connecting segments with nearby endpoints, would inevitably generate cycles or static behaviors confined to local regions, failing to provide meaningful state coverage for downstream planning.
>
> In contrast, SCoTS generates trajectories that are both temporally extended and diverse through a principled mechanism. We achieve temporal extension by guiding the stitching process along consistent directions within a learned temporal distance-preserving latent space. Diversity is ensured by assigning a unique, randomly sampled latent direction to guide the generation of each augmented trajectory. This principled strategy produces augmented datasets containing meaningful state-covering trajectories beneficial for training both diffusion planners and general GCRL methods.
>
> - **In the first paragraph of Section 3.2, there seems to be no difference between the unit vector $\boldsymbol{z}$ and the vector itself in $\boldsymbol{z}\gets \boldsymbol{z}/\|\boldsymbol{z}\|$.**
>
> Thank you for pointing out this. The expression $\boldsymbol{z}\sim\mathcal{N}(\mathbf{0},\mathbf{I})\,\boldsymbol{z}\gets \boldsymbol{z}/\|\boldsymbol{z}\|$ is intended to describe a two-step procedure: 1) a vector is first sampled from a standard Gaussian distribution, and 2) this vector is then normalized to become a unit vector. To enhance clarity in the final version, we will revise the text to explicitly separate these two steps,
>
> - **The state encoder $\phi(s)$ should take the state as input, but in Eq. (8), $\phi(\mathcal{D})$ appears to take the dataset $\mathcal{D}$ as input.**
>
> **$\phi(\mathcal{D})$** is a concise representation for the set of latent embeddings of the initial states of all trajectories within the dataset **$\mathcal{D}$**. This set forms the search space for candidate segments. We will clarify this shorthand in the final version to improve readability.
>
> References:
>
> [1] Yoon et al., Monte Carlo Tree Diffusion for System 2 Planning. ICML, 2025.
>
> [2] Chen et al., Simple Hierarchical Planning with Diffusion. ICLR, 2024.
>
> [3] Park et al., Lipschitz-constrained Unsupervised Skill Discovery. ICLR, 2022.
>
> [4] Park et al., METRA: Scalable Unsupervised RL with Metric-Aware Abstraction. ICLR, 2024.
>
> [5] Bae et al., TLDR: Unsupervised Goal-Conditioned RL via Temporal Distance-Aware Representations. CoRL, 2024.

---

> > ### Comment · Reviewer_C6Zi · 2025-08-05
> >
> > I thank the authors for their responses to my questions. They have mostly addressed my concerns. However, I still have some questions regarding the method. As far as I know, some prior works on trajectory stitching in diffusion planning, e.g., CompDiffuser [1], have already applied the state-covering trick. Other than that, what are the main contributions compared to these prior works, such as CompDiffuser [1], DiffStitch [2], and TS [3]? Are the primary differences the extension to a goal-conditioned scenario and the use of KNN?
> >
> > [1] Luo, Yunhao, et al. "Generative Trajectory Stitching through Diffusion Composition." arXiv preprint arXiv:2503.05153, 2025.
> >
> > [2] Li, Guanghe, et al. "DiffStitch: Boosting Offline Reinforcement Learning with Diffusion-based Trajectory Stitching." Proceedings of the 41st International Conference on Machine Learning, 2024.
> >
> > [3] Hepburn, Charles Alexander, and Giovanni Montana. "Model-based Trajectory Stitching for Improved Offline Reinforcement Learning." 3rd Offline RL Workshop: Offline RL as a "Launchpad", 2022.

---

> > > ### Author Response · Authors · 2025-08-05
> > >
> > > We sincerely thank reviewer C6Zi for thoroughly reviewing our responses. Your question about clearly distinguishing SCoTS from prior trajectory stitching methods such as CompDiffuser [1], DiffStitch [2], and TS [3] is crucial, and we appreciate the opportunity to elaborate further.
> > >
> > > The primary contributions of SCoTS are more fundamental than an extension to goal-conditioned scenarios or the use of KNN for search. The novelty lies in what we augment and how we augment it.
> > >
> > > To clearly differentiate from CompDiffuser [1], which indeed serves as a baseline in our paper, the primary distinction lies in the stage and manner of trajectory stitching:
> > >
> > > - **CompDiffuser [1] performs implicit stitching during inference (test-time).** It employs a compositional generative model that concurrently denoises multiple overlapping trajectory chunks to produce coherent plans at inference.
> > > - In contrast, **SCoTS explicitly performs stitching as a dataset augmentation step before training.** Our framework systematically generates extended and diverse trajectories to augment the offline dataset. This proactive augmentation approach not only enhances diffusion planners but also significantly boosts the performance of traditional offline goal-conditioned RL methods, demonstrating broader applicability.
> > >
> > > Regarding **DiffStitch [2]** and **TS [3]**, the distinction highlights another essential aspect of our contribution:
> > >
> > > - Both DiffStitch and TS fundamentally rely on **reward-driven** stitching. Their primary goal is optimizing trajectories by connecting low-reward segments to high-reward segments, inherently guided by a learned value or reward function. This reward dependency biases the augmented dataset toward specific tasks, limiting generalization across varied task scenarios.
> > > - Conversely, **SCoTS is intentionally designed to be reward-free.** Our primary objective is not to optimize for any specific reward function but rather to significantly enhance the dataset’s coverage and diversity. By focusing on directional and novelty-driven exploration, SCoTS produces extensive, diverse trajectories. Consequently, the augmented dataset is not biased toward a single reward landscape.
> > >
> > > Thank you again for your valuable feedback. Please let us know if there is anything else we can clarify.
> > >
> > > [1] Luo, Yunhao, et al. "Generative Trajectory Stitching through Diffusion Composition." arXiv preprint arXiv:2503.05153, 2025.
> > >
> > > [2] Li, Guanghe, et al. "DiffStitch: Boosting Offline Reinforcement Learning with Diffusion-based Trajectory Stitching." Proceedings of the 41st International Conference on Machine Learning, 2024.
> > >
> > > [3] Hepburn, Charles Alexander, and Giovanni Montana. "Model-based Trajectory Stitching for Improved Offline Reinforcement Learning." 3rd Offline RL Workshop: Offline RL as a "Launchpad", 2022.

---

> > > > ### Author Response · Authors · 2025-08-07
> > > >
> > > > Dear Reviewer C6Zi,
> > > >
> > > > Thank you again for the opportunity to further clarify the contribution of SCoTS. We have provided a detailed response to your question and hope it clarifies the key distinctions between SCoTS and prior works like CompDiffuser, DiffStitch, and TS. As the discussion period is nearing its end, we would greatly appreciate it if you could let us know if you have any remaining questions or suggestions.
> > > >
> > > > Best,
> > > >
> > > > The Authors

---

### Comment · Area_Chair_Lgt7 · 2025-08-06
**Question for the authors**

Hi,

Thanks a lot for preparing an informative rebuttal and engaging with the reviewers!
I have one question on my side: apart from a few additional experiments added in the rebuttal, the empirical validation is limited to navigation problems where the nature of the dynamics and the data are such that temporal-aware embeddings easily capture the variables that are more relevant to the tasks at hand (basically the x-y coordinates) and are semantically useful for stitching. I'm wondering how robust this approach is in other domains, eg, cube in OGBench, where the representation learning part should capture at least x-y-z to perform meaningful trajectory augmentation.

Thanks!

---

> ### Author Response · Authors · 2025-08-06
>
> Dear Area Chair,
>
> Thank you for your insightful question and for carefully reviewing our rebuttal. We appreciate you raising this important point regarding the generalizability of our trajectory augmentation method.
>
> We believe the temporal-aware embedding employed by SCoTS is robust and generalizable to other domains. Specifically, even in high-dimensional state spaces, such as AntMaze (29 dimensions), leanred temporal-aware embedding successfully captures relevant temporal and spatial aspects beyond simple x-y coordinates. For instance, in AntMaze, the learned embedding effectively distinguishes states not only by spatial position but also by orientation and posture.
>
> Nevertheless, we acknowledge that tasks involving precise object manipulation, such as controlling the cube in OGBench, may require additional considerations. For example, even minor discrepancies in the cube's position can imply large temporal distances, significantly limiting the availability of suitable stitching candidates. To handle this, we could consider a hindsight-inspired candidate generation step: when searching for candidate segments, we can artificially adjust cube states from other trajectories that fall within a small epsilon-distance from the current query trajectory's cube state. Specifically, we would relabel these nearby states to exactly match the cube position of the current query trajectory, effectively enlarging the candidate set and facilitating more robust trajectory stitching.
>
> In summary, while the locomotion tasks provide a clear and intuitive validation, the principles of SCoTS are general. We believe that with a principled modification, such as the proposed hindsight-inspired state matching, the framework can be effectively extended to object-centric domains, which we leave for future work.
>
> Best regards,
>
> The Authors

---

> > ### Comment · Area_Chair_Lgt7 · 2025-08-08
> > **Discussion**
> >
> > Thanks!

---

### Note · Authors · 2025-08-12

We appreciate the thoughtful feedback and constructive discussions from the reviewers and AC, which have helped us strengthen the clarity and contributions of this work.

SCoTS, a reward-free trajectory augmentation method distinct from prior works that perform test-time stitching or rely on task-specific reward signals to guide stitching, not only enhances diffusion planners but also significantly boosts the performance of existing offline goal-conditioned RL methods, demonstrating its broader applicability.

Methodologically, we clarified how balancing directional progress with novelty counteracts imperfections in the learned temporal-distance embedding, and how the diffusion refinement step enhances dynamic consistency of stitched trajectories.

We expanded the empirical study following the reviewers’ suggestions by adding dense-reward D4RL benchmarks, testing SCoTS on a mobile-robot navigation task with noisy sensing and partial observability, and providing comprehensive ablation studies justifying each component and sensitivity analyses on key hyperparameters.

We will incorporate all clarifications and new results from the rebuttal into the camera-ready version. Thank you again for your time and dedication.

---

### Decision · Program_Chairs · 2025-09-17

**Decision:**

Accept (poster)

**Comment:**

The paper introduces a novel technique for trajectory augmentation to be used in conjunction with trajectory diffusion methods to generate a more diverse and covering dataset that can be used in offline RL algorithms (e.g., goal-condition RL or diffusion planner). The paper illustrates the effectiveness of the proposed method in a few OGBench domains and show a non-trivial improvement across offline RL algorithms and initial datasets.

Trajectory datasets tend to provide a limited level of coverage as they are mostly composed by "expert" demonstrations and any "downstream" method  (BC/offline RL/diffusion-based planners) suffer from this limitations in terms of the breadth of problems that can be effectively solved. Finding ways to reliably expand the coverage and diversity of the initial dataset (e.g., by stitching trajectories) has the potential of directly improving the performance of many different RL algorithms. As such, there was clear consensus about the significance of the problem studied in this paper. Nonetheless, in the first round of reviews, the reviewers raised concerns on the proposed method and the empirical validation. After the rebuttal, all reviewers agreed on the fact that the clarifications provided by the authors and the additional empirical evidence on the effectiveness of the methods across other domains successfully resolved their initial concerns. As a result, there is now a general agreement towards accepting the paper.

I encourage the authors to improve their initial submission on these aspects:
* Integrate the new results: the precise implementation of the proposed method in the initial experiments may suggest that its effectiveness was tightly connected with the specific domain (Ant/PointMaze navigation). Showing that it is more broadly applicable and effective was very important to support the generality of the method.
* Include the discussion on the additional references suggested the authors.
* As the proposed method includes several components, it is crucial to include the ablations from the rebuttal to provide more support to how and why each aspect of the algorithm contributes to the final results.